# Donor-strand exchange drives assembly of the TasA scaffold in *Bacillus subtilis* biofilms

Jan Böhning [1], Mnar Ghrayeb[2,3], Conrado Pedebos[4], Daniel K. Abbas[1], Syma Khalid [4], Liraz Chai [2,3] ✉ & Tanmay A. M. Bharat [1,5] ✉

Many bacteria in nature exist in multicellular communities termed biofilms, where cells are embedded in an extracellular matrix that provides rigidity to the biofilm and protects cells from chemical and mechanical stresses. In the Gram-positive model bacterium *Bacillus subtilis*, TasA is the major protein component of the biofilm matrix, where it has been reported to form functional amyloid fibres contributing to biofilm structure and stability. Here, we present electron cryomicroscopy structures of TasA fibres, which show that, rather than forming amyloid fibrils, TasA monomers assemble into fibres through donor-strand exchange, with each subunit donating a β-strand to complete the fold of the next subunit along the fibre. Combining electron cryotomography, atomic force microscopy, and mutational studies, we show how TasA fibres congregate in three dimensions to form abundant fibre bundles that are essential for *B. subtilis* biofilm formation. Our study explains the previously observed biochemical properties of TasA and shows how a bacterial extracellular globular protein can assemble from monomers into β-sheet-rich fibres, and how such fibres assemble into bundles in biofilms.

Biofilms are the primary mode of microbial multicellular existence in nature[1]. Biofilms form in environmental settings as well as inside the bodies of living organisms during infections, and are further commonly found on abiotic surfaces such as medical devices[2]. In biofilms, bacterial cells encase themselves in a matrix of extracellular polymeric substance, made primarily of filamentous polymeric molecules, including proteins, polysaccharides, and DNA[3]. The biofilm matrix protects cells against physical and chemical stresses including antibiotic treatment[1–5] and is a hallmark of all bacterial biofilms. Therefore, understanding how polymeric molecules assemble in the extracellular matrix (ECM), and how they serve to scaffold biofilms, is of fundamental importance. Nevertheless, to the best of our knowledge, no high-resolution structures of biofilm scaffold fibres are available to help bridge this mechanistic gap in our understanding of biofilm formation.

The soil bacterium *Bacillus subtilis* is one of the best-studied model organisms for investigating biofilm formation[6–12]. The *B. subtilis* biofilm ECM is rich in exopolysaccharide (EPS) and protein components, the major proteinaceous component being a 26 kDa fibre-forming protein called TasA[13,14]. TasA is expressed as part of the *tapA-sipW-tasA* operon, which is controlled by the biofilm regulator protein SinR[15,16]. TasA was originally characterised as a spore coat-associated protein with antimicrobial activity, deletion of which results in defects in spore coat structure[14]. Later studies revealed that TasA assembles into fibres that form the major component of the *B. subtilis* biofilm matrix[13], with deletions of *tasA* resulting in significant defects in biofilm and pellicle formation that can be rescued by addition of exogenous TasA[17]. The *tasA* operon further encodes two accessory proteins; TapA, a minor matrix component often found associated with TasA, which is thought to be involved in anchoring TasA to the cell wall and in

[1]Sir William Dunn School of Pathology, University of Oxford, Oxford OX1 3RE, UK. [2]Institute of Chemistry, The Hebrew University of Jerusalem, Edmond J. Safra Campus, Jerusalem 91904, Israel. [3]The Center for Nanoscience and Nanotechnology, The Hebrew University of Jerusalem, Edmond J. Safra Campus, Jerusalem 91904, Israel. [4]Department of Biochemistry, University of Oxford, Oxford OX1 3QU, UK. [5]Structural Studies Division, MRC Laboratory of Molecular Biology, Francis Crick Avenue, Cambridge CB2 0QH, UK. ✉e-mail: liraz.chai@mail.huji.ac.il; tbharat@mrc-lmb.cam.ac.uk

accelerating its assembly[18,19], and the membrane-associated signal peptidase SipW that cleaves off the signal peptides of TasA and TapA[20].

An X-ray atomic structure of monomeric TasA has been solved showing a globular subunit, with a classical Jellyroll fold[21]. Interestingly, TasA fibres have been shown to possess several properties characteristic of amyloids, including high β-sheet content, resistance to depolymerisation and mechanical disruption, binding to the amyloid-specific A11 antibody, and staining positive in Congo Red and Thioflavin T assays[13,17,21–23], suggesting that TasA belongs to a growing list of so-called functional amyloid fibres[24], which also include biofilm matrix proteins in several Gram-negative bacteria such as *Escherichia coli*[25] and *Pseudomonas fluorescens*[26]. In contrast, a recent study proposed that TasA fibres consist of a linear arrangement of globular subunits that does not involve structural rearrangements in the transition of TasA monomers to fibres, suggesting that fibres formed by recombinant TasA are not amyloid[27]. Moreover, a recent X-ray diffraction study showed that native TasA fibrils only produce a weak cross-β-sheet pattern in vitro and in situ[11]. Given a lack of direct structural data, it is not possible to relate these previous observations to the underlying arrangement of TasA.

In this study, we address these ambiguities and bridge the gap in our mechanistic understanding of how abundant biofilm matrix proteins scaffold biofilms using electron cryomicroscopy (cryo-EM) structure determination, electron cryotomography (cryo-ET) and atomic force microscopy (AFM) of purified single fibres, reconstituted fibre bundles, and biofilms. Combining our structural data with molecular dynamics (MD) simulations and mutational studies, we provide a structural basis of biofilm scaffolding by the TasA protein in the Gram-positive model organism *B. subtilis*.

## Results

### Atomic structure of TasA fibres

To gain insights into the structure of TasA fibres, we purified TasA from a *B. subtilis* double mutant strain (Δ*sinR* Δ*eps*) that lacks the matrix polysaccharide operon *eps*, thereby producing TasA that is not associated with EPS, as well as the regulator gene *sinR*, an *eps* and *tasA* repressor, causing the strain to overproduce TasA[17]. Following cryo-EM sample preparation, we observed highly ordered fibres with a characteristic periodic appearance on cryo-EM grids (Fig. 1a), consistent with previous EM studies[27,28]. We performed helical reconstruction

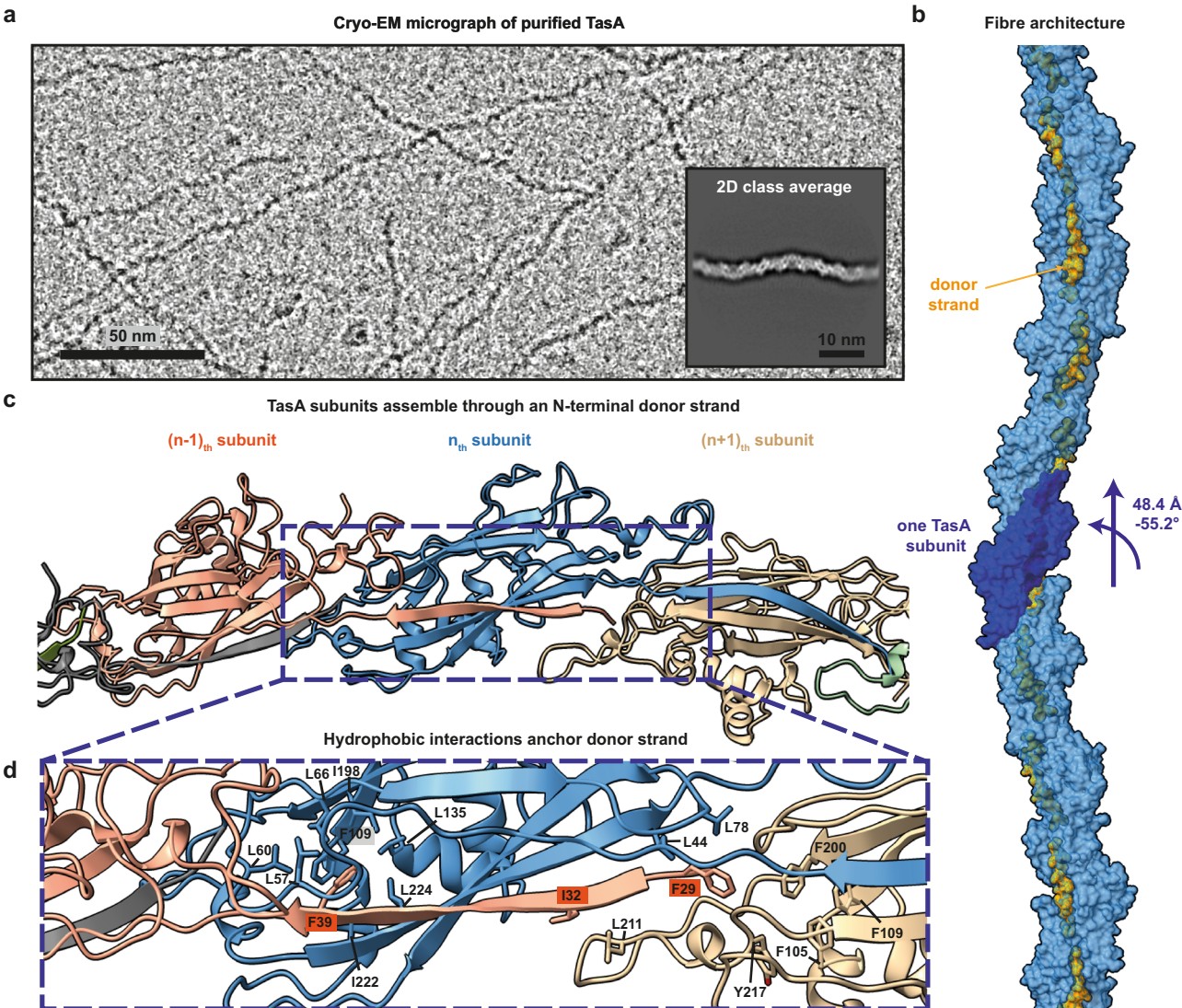

**Fig. 1 | TasA polymerises through donor-strand exchange. a** Cryo-EM image of TasA purified from *B. subtilis* Δ*sinR* Δ*eps* shows fibres with a characteristic periodic appearance. The inset shows a two-dimensional class average of TasA fibres. **b** Arrangement of TasA subunits in fibres. One subunit is highlighted in a darker shade of blue, with the N-terminal donor strands in orange. **c** TasA subunits polymerise through an N-terminal donor strand completing the β-sandwich fold of the following (n + 1)th subunit. **d** Hydrophobic residues in the (n-1)th subunit mediate donor strand interactions with the (n)th and (n + 1)th subunits.

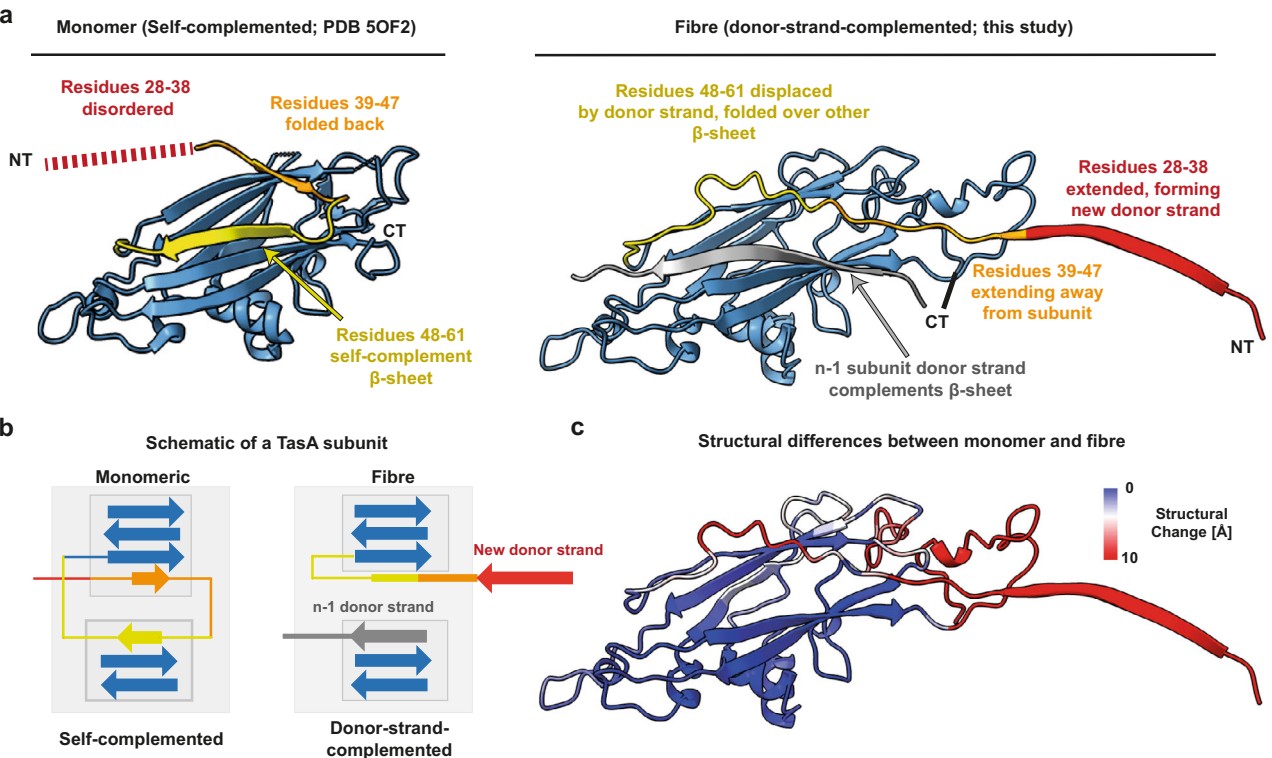

**Fig. 2 | TasA monomers undergo re-arrangement at the N- and C-termini upon fibre formation. a** Structural comparison of soluble, monomeric TasA (PDB 5OF2 [https://doi.org/10.2210/pdb5OF2/pdb], left) and fibrous TasA (right). Segments of the N-terminus are coloured in yellow (residues 48–61), orange (residues 39–47) and red (residues 28–38) to facilitate visualisation. The C-terminus (residues 239–261) in monomeric, self-complemented TasA (PDB 5OF2 [https://doi.org/10.2210/pdb5OF2/pdb]) is unstructured and thus not shown. C-terminal (CT) and N-terminal (NT) ends of the protein chain are indicated. **b** Schematic of β-sheet architecture (not to scale) demonstrating the structural differences between monomeric and fibrous TasA. Same colours as in **a** are used. **c** Cryo-EM structure of TasA (fibre-form) coloured based on structural deviations from the monomeric form. Large deviations are observed at the N- and C-termini. Regions unstructured in monomeric TasA are coloured red.

from cryo-EM images and determined a 3.5 Å resolution structure of TasA fibres (Figs. 1b–d and S1–2; Movie S1; Table S1; Methods). Interestingly, this structure shows that TasA monomers polymerise into fibres through an N-terminal β-strand that extends from each subunit into the next (Fig. 1b–d; Movie S1). There, this extended 'donor strand' completes a β-sheet of the following $(n + 1)_{th}$ subunit, and in this way provides an extensive inter-subunit interaction. In addition to the β-sheet hydrogen bonding of the donor strand with the next subunit, there are extensive hydrophobic interactions between donor strand residues (F29, I32, F39) and the next two subunits (Figs. 1d and S2). The C-terminal residues 239–261, previously found to be unstructured in NMR studies of monomeric TasA and not present in the monomer X-ray structure[21], are ordered in the fibre structure and form another inter-subunit interface (Fig. S2b). Thus, the fibre structure is neither representative of cross-β amyloids, nor is it a head-to-tail arrangement of globular subunits as previously suggested, but shares features of both, including inter-subunit interactions mediated by β-strands and an intact core fold of the monomer. The substantial extent of inter-actions between TasA subunits is likely responsible for the significant stability of the fibres previously observed[17], and the β-sheet-rich architecture and extensive β-strand interactions between subunits may offer an explanation for why, even though TasA fibres do not form an amyloid structure[29], they show several characteristics of amyloids[17,19]. Interestingly, a density in the cryo-EM map that may represent a cation, coordinated by two negatively charged aspartate (Asp) residues, was observed (Fig. S2c), which may be an ancestral remnant of the camelysin family of metalloproteinases, from which TasA has been suggested to derive[21].

Remarkably, the subunit interactions through a donor β-strand observed in TasA fibres are reminiscent of Gram-negative bacterial pilus assembly systems (Type I and V pili, Fig. S3a) that are based on donor-strand exchange[30,31]. Interestingly, a β-sandwich fold is not only observed in such pilins, but also in non-bacterial proteins undergoing β-strand exchange, such as the Mcg IgG protein involved in light-chain amyloidosis (Fig. S3b)[32] and uromodulin[33], which together suggest that a β-sandwich fold may be especially amenable for facilitating donor-strand complementation for the formation of fibrous assemblies.

## Rearrangement of N- and C- termini drives TasA fibre formation

The donor β-strand shields a significant hydrophobic groove in the fold of TasA (Fig. S4a) that would otherwise be exposed if not complemented by a donor strand. It is, however, known that the monomeric form of TasA protein is soluble and stable[21,34], indicating that residues are differently arranged between the monomeric and fibrous form of TasA. In order to analyse the structural differences in these two forms of TasA, we compared the monomeric, soluble form of TasA (PDB 5OF2 [https://doi.org/10.2210/pdb5OF2/pdb]) with the fibre form solved in this study (Fig. 2a, b; Movie S2). Remarkably, we observed that the binding site of the donor strand is self-complemented by a different β-strand (residues 48–61) in the monomeric form (Fig. 2a, b). In the fibre form, the previously self-complementing β-strand (residues 48–61) is displaced by the donor strand of the $(n-1)_{th}$ subunit (residues 28–38). The previously self-complementing strand then folds over the opposite β-sheet, and the donor strand (residues 28–38), along with residues 39–47, extends towards the next $(n+1)_{th}$ subunit, allowing assembly of the next subunit (Fig. 2a). The C-terminus of TasA is well-

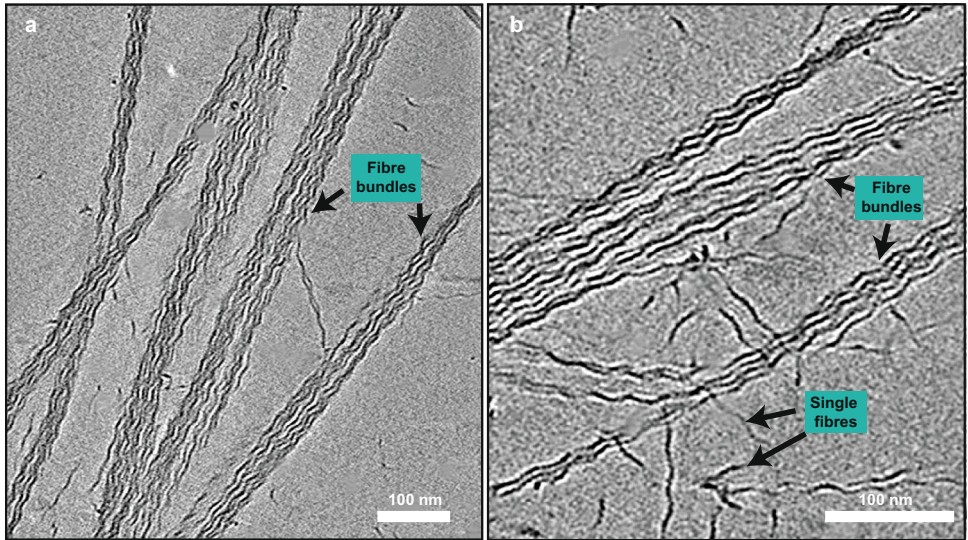

**Fig. 3 | TasA fibres spontaneously associate into bundles. a, b** Cryo-ET slices of purified TasA show the formation of locally ordered fibre bundles. Both bundles and single fibres can be seen in the specimen (marked with arrows).

resolved in the fibre structure (Figs. 2a–c and S2b), suggesting a role of the C-terminal residues in promoting fibre stability.

Our structure suggests that, unless a TasA subunit is donor-strand-complemented, it cannot extend its own donor strand as this would otherwise expose a hydrophobic groove (Fig. S4a). Previous studies have shown that the accessory protein TapA nucleates TasA fibre formation in native cellular systems[19], and that it accelerates the assembly of TasA into fibres in vitro[18]. We noticed significant sequence similarity between the TapA N-terminal residues and the TasA donor strand (Fig. S4b), suggesting TapA might extend a donor strand itself. Since small, sub-stoichiometric subunits of protein fibre tips near the cell surface are not amenable to helical cryo-EM reconstruction, we employed AlphaFold-Multimer[35] to predict whether TapA could complement a TasA subunit. Indeed, the structural model shows the N-terminal TapA residues extending into the β-sandwich domain of TasA, completing the fold (Fig. S4b). Together with previous results showing that TapA nucleates fibre formation[18,19], this suggests that TapA may promote the donor-strand exchange reaction by providing the first donor strand during fibre formation in biofilms. This also provides a structural explanation for a previous observation that only a fraction of N-terminal residues of TapA are required for biofilm structuring by TasA[36].

### Interactions between TasA fibres

To obtain a structural understanding of how TasA fibres form the bundles observed in biofilms[22], we performed cryo-ET analysis on TasA bundles reconstituted in vitro (Fig. 3; Movie S3). While we observed TasA fibres associated into large bundles, smaller bundles and even single fibres could consistently be detected, implying a low-affinity interaction between fibres. This observation is consistent with previous studies showing that TasA bundling is concentration-dependent[28], possibly relying on avidity effects for assembly. TasA fibres within the bundle were orientationally aligned (Fig. 3), but we were not able to detect distinct, ordered fibre-fibre interactions in Fourier transforms of bundles or through subtomogram averaging (STA) of fibres within bundles (Fig. S5).

Interestingly, our cryo-ET data contains some doublets of TasA fibres, which represent the simplest interaction between fibres (Fig. 4a–c). A basic overlay of our cryo-EM model with the density of the TasA doublet shows that fibres do not interact consistently all along the length of the doublet, but rather periodically every ~32 nm. Geometrical considerations show a loop containing residues 174–177

as the structural element that extends furthest out from the TasA fibre, ideally placed for mediating fibre-fibre interactions (Fig. 4d).

To determine whether fibre-fibre interactions are mediated through this part of the protein, we performed molecular dynamics (MD) simulations on a two-fibre system (Fig. S6; Movie S4) with both parallel and antiparallel fibre orientations. The simulations confirmed a stable interaction that includes both the extended loop region (residues 173–179, IDGKTAP) as well as some additional residues at the fibre-fibre interface (Fig. S6). To further verify our structural observations, we generated a genomic *B. subtilis* mutant expressing TasA in which residues 174–177 are mutated to poly-alanine (174–177$_{AAAA}$), which, according to our model, would disrupt fibre-fibre bundling. Colony biofilms and pellicles with this mutation (174–177$_{AAAA}$) showed an aberrant morphology compared to wild-type *B. subtilis* biofilms, reminiscent of a Δ*tasA* strain (Figs. 5a–c and S7a), indicating that the bundling ability of TasA had been disrupted, supporting the inferences from our cryo-EM data and MD-based doublet model. Also, cryo-EM of the 174–177$_{AAAA}$ mutant fibres showed markedly reduced bundling (Fig. S7c, d, 24% wild-type versus 2% for the mutant); however, this correlated with a smaller amount of TasA fibres formed by the mutant, as indicated by less fibres being detectable in the sample overall, despite the use of the same concentration of TasA.

Additionally, we generated mutant strains expressing donor-strand-truncated TasA (Δ28–38), which form significantly defective biofilms and pellicles that show an aberrant morphology compared to wild-type *B. subtilis* (Figs. 5d and S7). Cryo-EM of Δ28–38 TasA fibres showed defective fibre formation in TasA isolates (Fig. S7), consistent with the role of the donor strand in fibre assembly suggested by our structural data.

Next, by imaging intact wild-type pellicle biofilms using atomic force microscopy (AFM), we observed networks of filaments adherent to cells, potentially formed by TasA fibres, that are severely reduced in Δ*tasA* biofilms (Figs. 5e, f and S8). Similarly, greatly increased numbers of fibres could again be seen in a strain that overproduces TasA but lacks EPS (Δ*sinR* Δ*eps*) (Fig. S8e, f). This further confirms the role of TasA in scaffolding biofilms by forming an integral part of the ECM.

We subsequently went on to test whether the fibres observed in biofilms correspond to those formed from purified TasA to rule out artefacts of in vitro preparation or a possible polymorphism of TasA fibres. Accordingly, we performed cryo-EM imaging of the ECM components of *B. subtilis* biofilms to examine the morphology of native TasA fibres. For this experiment, we used biofilms of an

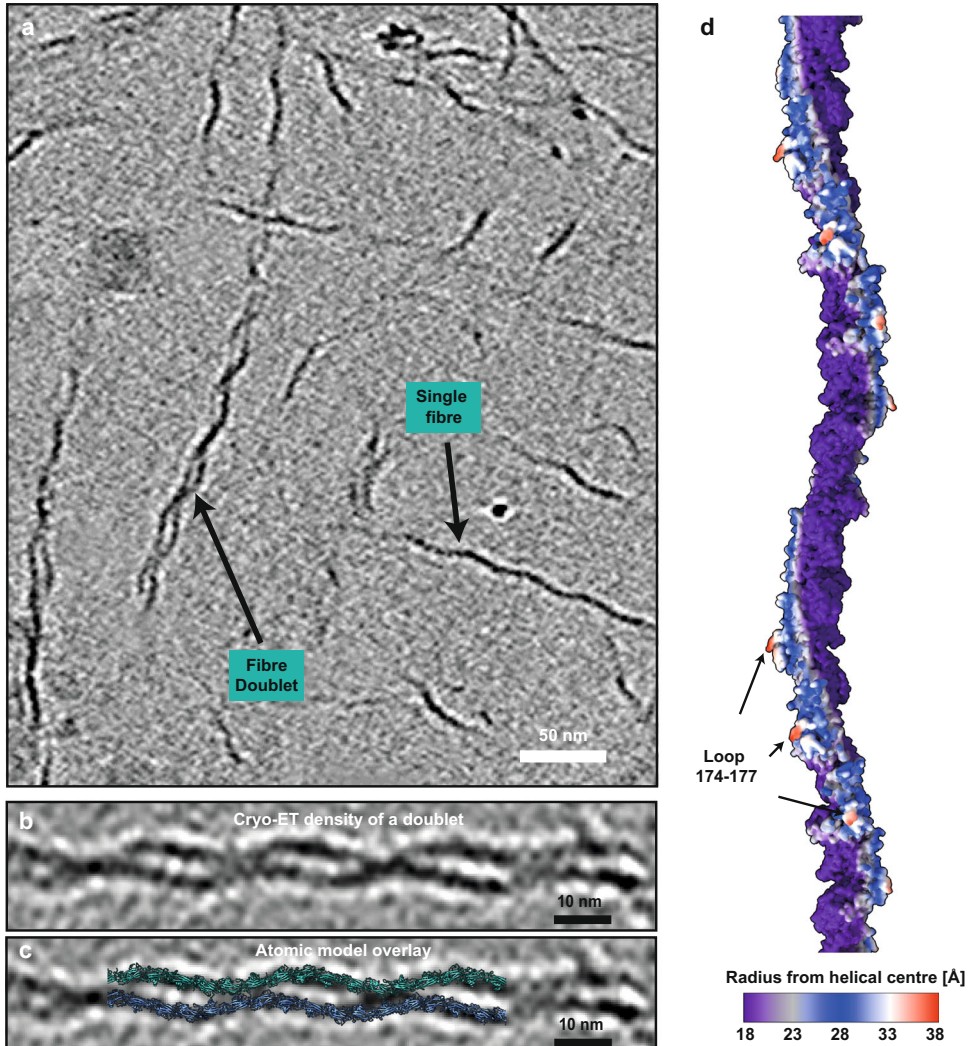

**Fig. 4 | TasA fibres form bundles through periodic interactions with neighbouring fibres mediated by distal loop residues. a** Tomographic slice showing TasA bundles, with the smallest bundles formed by two TasA fibres (marked with arrows). **b** Cryo-ET data depicting the interaction of two TasA fibres, revealing that fibre-fibre interactions are periodic. **c** Overlay of atomic models of two TasA fibres into the cryo-ET density. **d** Atomic model of a TasA fibre (18 subunits shown) coloured according to the radius from the helical centre (helical axis). The part of the filament extending out farthest (red) corresponds to the loop containing residues 174–177.

exopolysaccharide-deficient *B. subtilis* mutant (Δ*sinR* Δ*eps*), which, compared to wild-type *B. subtilis*, readily disassembled into its components upon gentle agitation and deposition on cryo-EM grids (Fig. 5g, h). A visual inspection of the specimen showed deposited *B. subtilis* cells together with components of the ECM (Fig. S9). We observed that a major component of the specimen consisted of TasA fibres, occurring predominantly in fibre bundles (Figs. 5g, h and S9), in line with previous reports[22]. Both fibre bundles and individual TasA fibres were observed on the grid, and strikingly, they were found to be identical to purified TasA (Fig. 5h, inset), including the same subunit arrangement and helical repeat, strongly suggesting that the donor-strand-exchanged TasA is the primary form of the protein in biofilms.

## Discussion

The atomic structure of TasA fibres shows how this protein retains characteristics of the monomeric protein[21], which would be expected in a linear arrangement of globular subunits[27,28]. At the same time, β-strand interactions between TasA subunits in the fibre result in a highly inter-connected and stabilized structure, as found in classical amyloids[13,17,21–23]. These unique features explain why previous studies have arrived at different expected subunit arrangements of TasA within the fibre. While it is well established that functional amyloids are abundant in bacteria[24], we propose that other suggested functional amyloids could also exhibit similar characteristics, and that there may be a continuum of fibre-forming proteins - on one end arranged as classical amyloids[25,37], to amyloid-like fibres displaying features shared with fibres formed from globular proteins on the other end[27].

The structure of TasA fibres reveals that the TasA system is a donor-strand exchange system in Gram-positive bacteria. While Gram-negative systems[30,31] do not appear to be evolutionarily related to the TasA/TapA system and employ different assembly systems, the widespread occurrence of donor-strand exchange suggests a common structural solution to the challenge of forming secreted protein fibres with high stability in the harsh and relatively uncontrolled extracellular space. Further studies into the TasA/TapA system will be needed to understand in detail the mechanisms of assembly of TasA fibres compared to Gram-negative pili, which employ Chaperone-Usher systems or proteolytic cleavage for assembly[30,31]. Interestingly, very recent studies have demonstrated that fibres in extremophile archaea also are donor-strand-complemented[38,39], with some archaeal systems even structurally related to TasA[39], which suggests that donor-strand

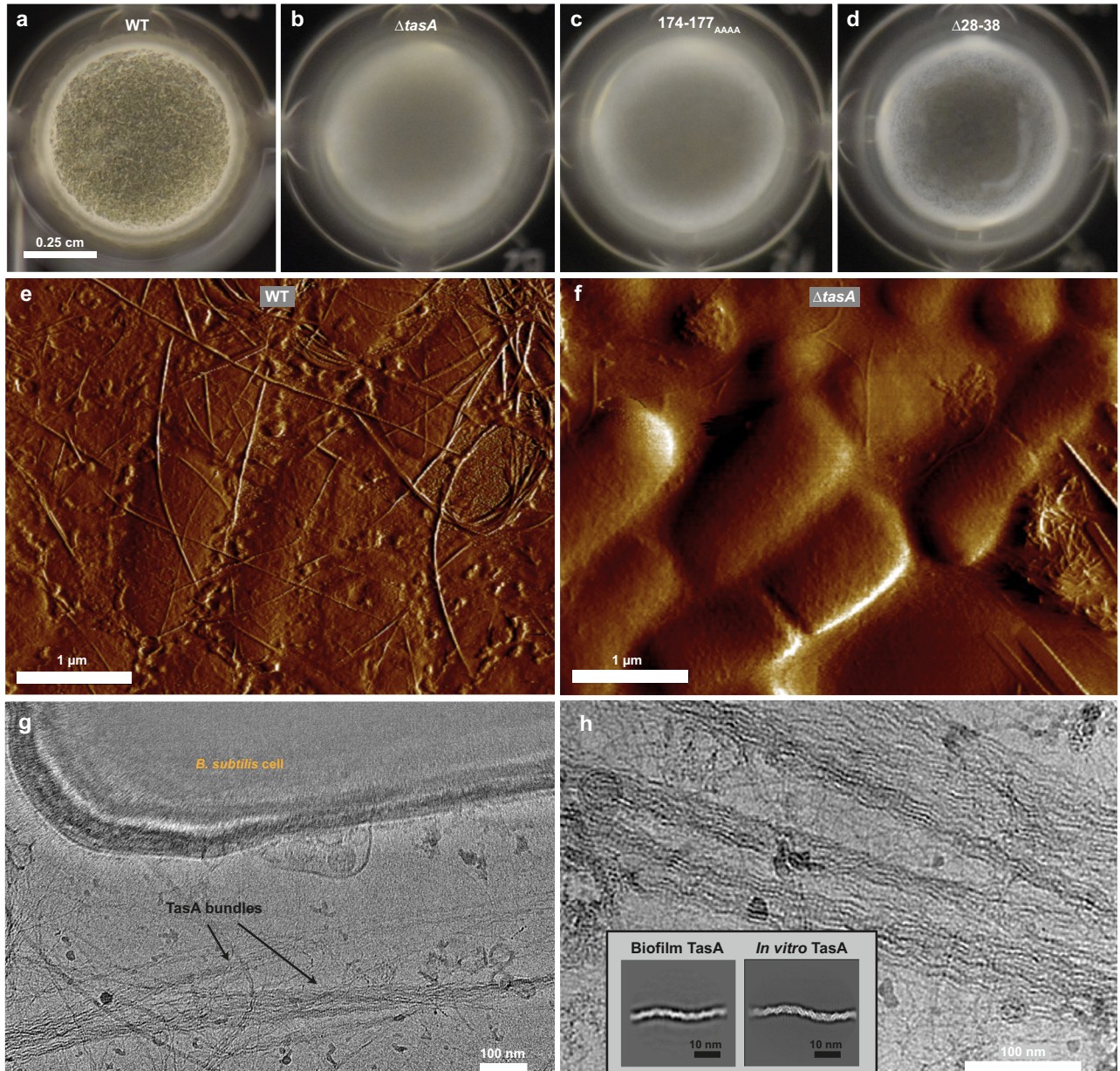

**Fig. 5 | TasA in *B. subtilis* biofilms.** 24-h old *B. subtilis* pellicles of strains: **a** NCIB 3610 wild-type, **b** Δ*tasA* (ZK3657 strain), **c** fibre-fibre interface mutant (174–177ₐₐₐₐ) *tasA* (MG4 strain) and **d** donor strand mutant (Δ28–38) *tasA* (MG3 strain, see Table S3 for strain list). AFM imaging comparing *B. subtilis* pellicles of **e** wild-type (ZK5041 strain, Table S3) and **f** Δ*tasA* (ZK3657 strain) shows that deletion of *tasA* results in a significant reduction in visible fibres. Fibres remaining in Δ*tasA* pellicles may correspond to flagella (see Fig. S9). **g** Cryo-EM image of the ECM components of a *B. subtilis* Δ*sinR* Δ*eps* (ZK4363 strain) biofilm showing bacterial cells along with TasA fibres and bundles. **h** TasA fibres in the ECM show the same structural features as purified TasA. Inset: Two-dimensional averaging shows that ECM biofilm and purified TasA are indistinguishable in their helical arrangements.

assembly is a ubiquitous strategy for the assembly of highly stable fibres.

In this study, we describe the first structure of a biofilm matrix fibre scaffold, enabling a structural understanding of the ECM of bacterial biofilms. In particular, we show that fibres interact through periodic interactions to stack onto each other to form three-dimensional bundles, scaffolding *B. subtilis* biofilms (Fig. 6). We were not able to resolve distinct interactions between fibres using STA (Fig. S5), which, in addition to lack of distinct lattice spots in Fourier transforms of bundles, suggests that interactions between fibres are not highly ordered. This interaction appears to be qualitatively similar to formation of tactoids by the phage Pf4 in *Pseudomonas aeruginosa* biofilms[40]. Nevertheless, our experiments show the importance of a

loop (containing residues 174–177) extending away from the fibre surface in mediating these interactions. Interestingly, in a recent study on a TasA-like system (archaeal bundling pili (ABP)) in hyperthermophilic archaea[39], fibres formed ordered, six-fibre bundles. We detected bundles of variable sizes for TasA, which also appear more flexible. Stronger interactions in ABP filaments may be required as the archaea containing ABP live in acidic and boiling waters. The exact mode of interaction between TasA fibres at the atomic structural level remains to be determined; however, our MD simulations suggest that both antiparallel and parallel TasA bundling could be possible.

Our mutational, cryo-EM and AFM experiments together confirm that donor-strand-exchanged TasA fibres form a large and crucial part of the *B. subtilis* biofilm ECM, and also confirm that TasA's role as a

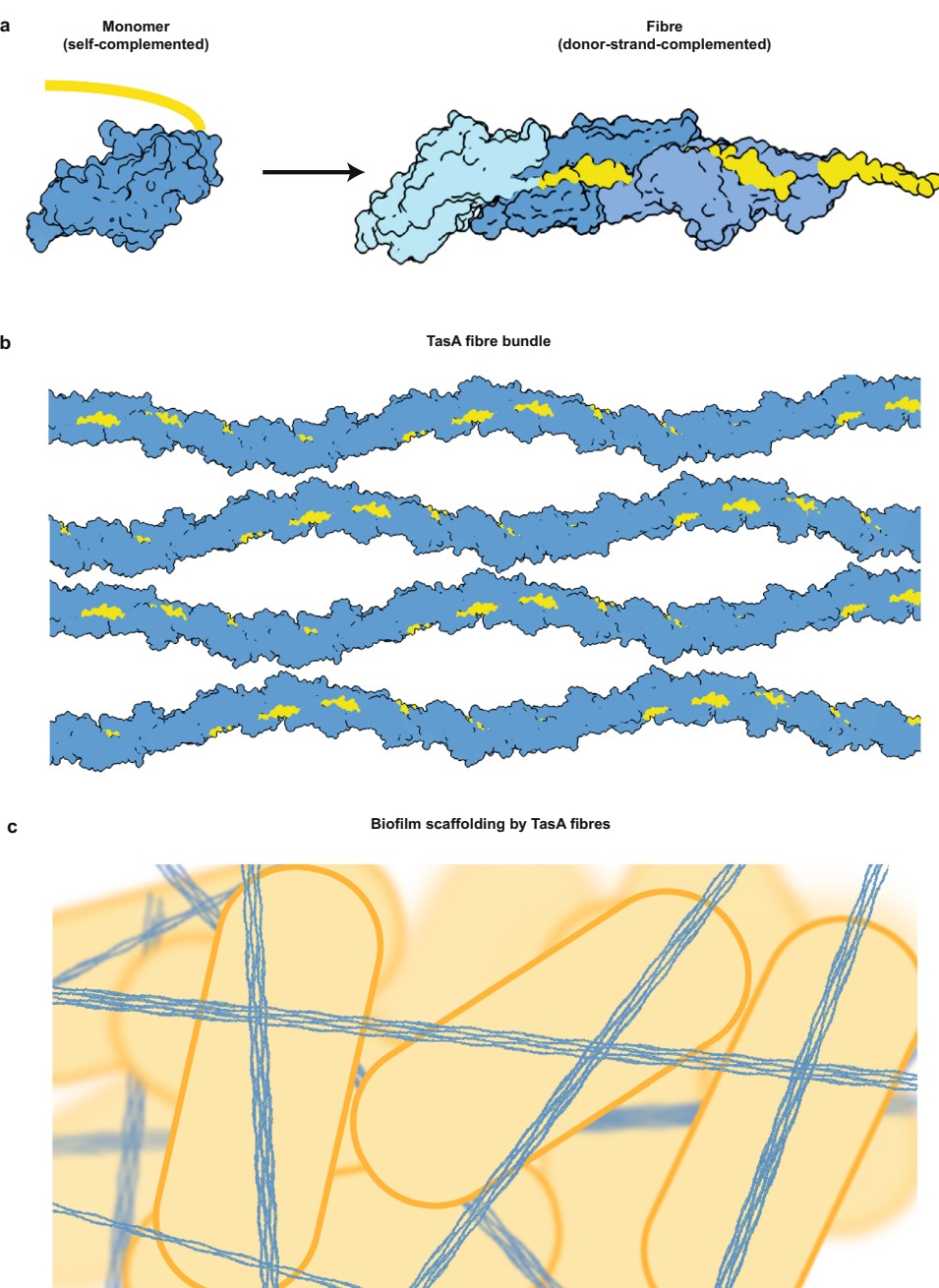

**Fig. 6 | Schematic model of biofilm scaffolding by TasA. a** Formation of fibres by transitioning from a self-complemented monomer to a donor-strand-complemented fibre. **b** Bundle formation through periodic interactions of TasA fibres. **c** AFM-based schematic of biofilm scaffolding through formation of TasA fibre networks.

scaffold is vital for the formation of rigid biofilms. This study thus establishes the structural basis of how filamentous fibres form a scaffold in the ECM of bacterial biofilms, with potential implications on other such systems including fibrous proteins that form bundles including curli from *E. coli*[25], Fap from *Pseudomonas*[26] and phenol-soluble modulins and Bap from *Staphylococcus aureus*[41,42].

A recurring theme in bacterial biofilms is the abundance of fila-mentous molecules including polysaccharides[43,44], proteins[24] and DNA[45] in the ECM[3], which may be important for the formation of microenvironments protecting cells within biofilms[40,46]. Further stu-dies into such mechanisms will be important to understand how the ECM is formed, which is of fundamental importance in comprehending how biofilms are built. Since the absence of the *B. subtilis* EPS facili-tated disassembly of the biofilm (Fig. 5g, h), in agreement with

previous work[27,47], this study provides another example of a synergy between filamentous proteins and ECM polysaccharides, also descri-bed in other model bacterial systems including pathogenic species such as *Vibrio cholerae* and *Pseudomonas aeruginosa*[43,44]. Taken toge-ther with previous studies, our data suggest that ordered, local inter-actions between filamentous molecules leading to the formation of superstructures in the ECM may be a general organisational principle for biofilm formation.

## Methods
### Construction of N-terminally truncated TasA (Δ28−38 TasA)
For the deletion of the 82–111 bp region of the *tasA* gene, regions upstream and downstream of the *tasA* gene were amplified using genomic DNA of wild-type *B. subtilis* PY79 strain. Polymerase Chain

Reaction (PCR) products T2838P1, T2838P2 (henceforth we refer to primers as P1, P2, and so on for simplicity; see Table S2 for full sequences) containing 1 Kb upstream of the *tasA* deletion region were amplified by the primers P1 and P2, and the PCR product P3P4 containing the downstream region of the *tasA* 82–111 bp deletion region up to the end of the *tasA* gene was amplified using the primers pair P3 and P4. An additional approximately 1 Kb region downstream to *tasA* was amplified using the primer pair P5 and P6. PCR reactions were performed using Q5 polymerase (New England Biolabs).

DNA fragments were purified using the nucleo-spin gel and PCR clean-up kit (Machery Nagel, cat# 740609.50). The assembly of DNA fragments (P1P2, P3P4, P5P6, and antibiotic genes) was performed using a Gibson assembly master mix kit (New England Biolabs, cat# E2611L). Assembled product was then transformed into PY79 and grown on selective (10 μg/ml) kanamycin agar plates to obtain the replacement with *tasA* Δ28–38. The mutated genomic DNA (gDNA) was extracted from PY79 using the Wizard genomic DNA purification kit (Promega, cat# A1620) following the manufacturer's instructions, and transformed into both wild-type 3610 and Δ*sinR* Δ*eps* (ZK4363, see Table S3) *B. subtilis* strains. Colonies were grown on kanamycin plates and mutated clones were sequence-verified using PseqF and PseqR. Primers (Integrated DNA Technologies, IDT) used in this study are listed in Table S2.

### Construction of mutated DGKT - > AAAA TasA (174–177$_{AAAA}$)
For the replacement of the DGKT (174–177)-encoding sequence of the *tasA* gene with a sequence encoding AAAA, yielding 174–177$_{AAAA}$, an approximately 1 Kb region upstream of the *tasA* region and a second approximately 1 Kb region downstream of *tasA* were amplified by PCR using the genomic DNA of wild-type *B. subtilis* PY79 strain as a template and two primer pairs T174AP1, T174AP2 and T2838P5, T2838P6. The PCR product of *tasA* 174–177$_{AAAA}$, was amplified using the primer pair T174AP3 and T2838P4. PCR products were treated as described above for *tasA* Δ28–38 for generating the mutant strain.

### Transformation protocol
gDNA (for either TasA Δ28–38 or 174–177$_{AAAA}$) was transformed into *B. subtilis* strains as follows. A single colony of PY79, NCIB 3610, or Δ*sinR* Δ*eps* (ZK4363*) B. subtilis* strains were picked and grown in 1XMC medium (10 mM potassium phosphate buffer pH 7.0, 0.3 mM sodium citrate, 0.2% (w/v) glucose, 2.2 mg/mL ferric ammonium citrate, 0.01% (w/v) casein hydrolysate, 0.02% (w/v) potassium glutamate) with 0.01 M MgSO$_4$ and incubated at 37 °C for 3.5 h (h) at 250 revolutions per minute (rpm). Then, 300 μl of the culture was mixed with 10 μl of the extracted gDNA (for either the TasA Δ28–38 or 174–177$_{AAAA}$) and incubated at 37 °C for additional 3 h at 250 rpm. The transformed cells were then plated on selective kanamycin agar plates and incubated at 37 °C for 16 h. Single colonies of each mutant were then picked and used for the TasA purification as described below.

### Native and mutated TasA purification
Native and mutated TasA were purified as described previously[17,34]. In brief, liquid cultures of the relevant *B. subtilis* strains, Δ*sinR* Δ*eps* (ZK4363), MG1, and MG2 (see Table S3) were grown for 16 h in LB (Luria Bertani) broth, diluted 1:100 into MSgg medium, and grown under shaking conditions (225 rpm) for 16 h at 37 °C. Cells were then pelleted by centrifugation (10,000 g, 15 min, 4 °C), the supernatant was discarded, and the cells were resuspended in saline extraction buffer[17], probe-sonicated (1518 J, 5 s pulse, and 2 s pulse off), and stirred for 30 min at room temperature. The supernatant was then collected after centrifugation (10,000 g, 15 min, 4 °C) and filtered through a 0.45 μm polyethersulfone (PES) bottle-top filter. The filtered supernatant was concentrated with GE Healthcare Vivaspin centrifugal filter tubes (10 kDa Molecular Weight Cut-off) and loaded on a HiLoad 26/60 Superdex S200 gel filtration column that was pre-equilibrated with a

20 mM Tris solution at pH 8 (for both TasA fibrils and bundles preparation). Purified protein was lyophilized and stored at −20 °C until used. TasA purification was verified using Sodium dodecyl sulphate-polyacrylamide gel electrophoresis (SDS-PAGE) (Fig. S7b) and their structure was characterized using circular dichroism (Fig. S7f). The concentration of TasA was determined using a BCA protein assay (ThermoFisher Scientific).

### Comparison of WT and mutated TasA protein expression using SDS-PAGE
The purification process described above was repeated using liquid cultures of *B. subtilis* strains, ZK4363, MG1, and MG2 (see Table S3). To make sure that the protein is purified from the same number of cells, liquid cultures were grown for 16 h in LB, and their optical density at 600 nm was adjusted to 1 using LB medium. Following a further 1:100 dilution into MSgg medium, cells were grown at 37 °C for 16 h under shaking conditions at 225 rpm. The OD of these cultures was adjusted to 1 using MSgg and cells were pelleted with centrifugation (10,000 g, 15 min, 4 °C). Protein was extracted from the pellet with sonication and stirring as described above. The supernatant was then filtered through a 0.45 μm polyethersulfone (PES) bottle-top filter and samples were loaded on an SDS-PAGE gel with ladder protein marker (GE Healthcare, cat# 17-0446-01) and stained with Coomassie Blue.

### TasA fibrils and bundles preparation
Native (or mutant) purified TasA preparation was passed through a HiLoad 26/60 Superdex S200 sizing column that was pre-equilibrated with 20 mM Tris (pH 8) and lyophilized. The protein was then dissolved with water to adjust the protein to 28 μM and the buffer concentrations to 25 mM Tris (pH 8) with 50 mM NaCl. The protein solutions were incubated at 4 °C for a minimum of two days to allow fibril and bundles formation.

### Circular dichroism
Circular dichroism (CD) spectra were recorded using a Jasco J-715 spectropolarimeter in the range of 190–260 nm using 1 mm quartz cuvettes. Five scans were recorded and averaged for each sample. CD spectra of buffer solutions were subtracted from the corresponding spectra and ellipticity (mº, in millidegrees) was converted to mean residue ellipticity (MRE) units using the following equation: $MRE = \frac{m^\circ * M_w}{10^4 * L * C * AA}$, where $M_w$ is the molecular weight of the protein in g/mol, L is the cuvette path length in cm, C is the protein concentration in mg/ml and AA in the number of amino acids in the protein sequence. CD was performed using 200 μl of TasA solutions (10 μM of TasA wild-type, 5 μM of Δ28–38 and 174–177$_{AAAA}$ mutants) in 10 mM potassium phosphate buffer, pH 8.0. CD spectra in Fig. S7 were plotted using Origin Pro 2022 SR1.

### Biofilm formation
Wild-type (NCIB 3610), *tasA* mutant ZK3657 (3610 tasA::kan) and mutant strains Δ*tasA*, MG3 (Δ28–38) and MG4 (*tasA* 174–177$_{AAAA}$) of *B. subtilis* were used for biofilm formation either on agar plates or as pellicles at the liquid-air interface, as specified below.

Biofilm formation on agar plates: Biofilms on agar plates were prepared by placing 2 μl of an overnight LB culture on a 1.5% (w/v) agar MSgg plates[6] and incubating at 30 °C for 3 days. *B. subtilis* Δ*sinR* Δ*eps* (ZK4363) biofilms for cryo-EM sample preparation (Fig. 5) were incubated at 30 °C for 5 days.

For pellicle formation at the liquid-air interface, pellicles were prepared in 96-well cell culture plates (Costar 3596) by a 1:50 dilution of an overnight LB culture into MSgg broth. Plates were then incubated at 30 °C for 24 h and pellicles were visualized using a Nikon D-5200 Camera with an 88 mm macro lens.

Biofilm and pellicle images presented in this study represent four biological repeats with at least three triplicates of each.

## Atomic force microscopy

Atomic force microscopy (AFM) topography scans were performed using a BioScope Resolve™ BioAFM (Bruker, Santa Barbara, CA, USA). Height sensor and Peak Force error AFM images were measured simultaneously in Peak Force Quantitative Nanomechanical Mapping (QNM) mode using ScanAsyst-Air probes (Bruker, resonance frequency 70 KHz, spring constant 0.4 N/m, tip nominal radius 2 nm). Peak force error images do not provide information on height and therefore they are presented without a height scale. However, with each peak force error image we also present the corresponding height sensor image in the extended data. Cantilever spring constants were calibrated using the Thermal Tune noise method.

Protein samples were adsorbed on mica as follows: 20 µl of 8 µM TasA in 20 mM Tris pH 8.0, 1.5 M NaCl was placed on a freshly cleaved mica and kept for 30 min in a humid atmosphere. The sample was then washed twice with water for 10 min and dried with a stream of nitrogen. AFM images of TasA fibers in Fig. S8c, d represent scans from at least 30 different areas of protein from at least six different purification batches.

Pellicle samples were prepared by collecting 24 h-old pellicles (ZK5041), resuspending them with 100 µl of 10 mM potassium phosphate buffer pH 8.0, and placing a 20 µl drop on petri dishes (WillCoDish, glass-bottom, Amsterdam) that were pre-incubated with poly-L-lysine. Samples were left to dry on the plate overnight. AFM images of WT biofilms (ZK5041) represent scans from three different biological repeats, and 26 different areas. AFM images of ΔtasA mutant biofilms (ZK3657) represent scans from two different biological repeats and eight different areas. AFM images of ΔsinR Δeps mutant biofilms (ZK4363) represent scans from two different biological repeats, and 10 different areas were scanned in total.

For poly-L-lysine adsorption on petri dishes, we incubated the petri dishes for 30 min with a 0.1 mg/ml poly-L-lysine solution in a humid atmosphere and discarded the poly-L-lysine solution prior to placing a pellicle drop.

## Cryo-EM and cryo-ET sample preparation

For cryo-EM grid preparation of purified TasA fibres, fibre isolates were diluted to final concentrations of 10 µM TasA in 12.5 mM Tris pH 8 and 25 mM NaCl, and 2.5 µl were applied to a freshly glow-discharged Quantifoil R 2/2 Cu/Rh 200 mesh grid and plunge-frozen into liquid ethane using a Vitrobot Mark IV (ThermoFisher) at 100% humidity at 10 °C. For tomography samples of purified TasA, 10 nm Protein-A-gold beads (CMC Utrecht) were added as fiducials prior to plunge-freezing. For cellular cryo-EM sample preparation, B. subtilis ΔsinR Δeps biofilm was scraped from a plate and resuspended in PBS immediately prior to sample preparation in the same manner.

## Cryo-EM and cryo-ET data collection

Cryo-EM data was collected on a Titan Krios G3 microscope (ThermoFisher) operating at an acceleration voltage of 300 kV, fitted with a Quantum energy filter (slit width 20 eV) and a K3 direct electron detector (Gatan). Images were collected in super-resolution counting mode using a pixel size of 1.092 Å/pixel for helical reconstruction of TasA fibres and 3.402 Å/pixel for imaging of biofilm and mutant TasA samples. For helical reconstruction of TasA, movies were collected as 40 frames, with a total dose of 48.5 electrons/Å², using a range of defoci between −1 and −2 µm. For imaging of B. subtilis biofilm ECM components, a total dose of 47.5 e/Å² distributed over 80 frames and defoci between −3 to −7 µm were used. Cryo-ET tilt series of TasA bundles were collected on the same microscope using SerialEM[48], with a total dose of 120–180 e/Å² and defoci of −8 µm, collected between ±60° tilt of the stage at 1° tilt increments.

## Cryo-EM data processing

Helical reconstruction of TasA fibres was performed in RELION[49,50]. Movies were corrected using the RELION 3.1 implementation of

MotionCor2[51], and CTF parameters were estimated using CTFFIND4[52]. Initial helical symmetry of TasA filaments was estimated from the observed periodicity of subunits within individual two-dimensional class averages. Initial models of TasA fibres were created from class averages using the relion_helix_inimodel2d programme. Three-dimensional classification was used to identify a subset of particles that supported refinement to 3.5 Å resolution. For final refinement, CTF multiplication was used for the final polished set of particles. Symmetry searches were used during reconstruction, resulting in a final rise of 48.36 Å and a left-handed twist per subunit of 55.2°. Resolution was estimated using the gold-standard FSC method as implemented in RELION. Local resolution measurements were also performed using RELION.

For STA, we employed the helical reconstruction method in RELION-4 with the previously determined helical parameters[53].

## Model building and refinement

Manual model building was performed in Coot[54]. The previously published crystal structure of the soluble monomer was used as a starting model (PDB 5OF1 [https://doi.org/10.2210/pdb5OF1/pdb]). Residues of the starting model that were inconsistent with the density of fibrous TasA, including loops and the N-terminal donor strand, were deleted and manually rebuilt. The initial model of fibrous TasA was subjected to real-space refinement against the cryo-EM map within the Phenix package[55]. Three subunits of TasA were built and used for final refinement. Non-crystallographic symmetry between individual TasA subunits was applied for all refinement runs. Model validation including map-vs-model resolution estimation was performed in Phenix.

## Tomogram reconstruction and fitting of atomic models

Tilt series alignment via tracking of gold fiducials was performed using the etomo package as implemented in IMOD[56]. Tomograms were reconstructed with WBP in IMOD[56] or SIRT in tomo3d[57]. Atomic models of TasA fibres containing eighteen subunits were fitted in parallel into the tomographic density using ChimeraX[58]. Fits consistently placed a loop containing residues 174–177, which faces outwards from the helical centre, at the inter-subunit interface, independent of fibre orientation.

## Molecular modelling and molecular dynamics simulations

Cryo-EM models of parallel and antiparallel TasA filament doublets were each submitted to the Martini Maker tool[59] from the CHARMM-GUI server[60,61] to build the initial coarse-grained model of the system. Each system was solvated with explicit water, and a number of counter ions were added to neutralize charges, with an extra salt concentration of 1 M of KCl ions for all simulations. The final systems obtained were composed of approximately 2.5 million particles each. Molecular dynamics simulations were performed using the GROMACS simulation suite (version 2020.4)[62] along with the Martini2.2 force field[63]. The ElNeDyn model with an elastic network cut-off of 0.9 nm and a force constant of 500 kJ.mol⁻¹.nm⁻² was used to coarse-grain the protein models[64]. Starting positions for the filaments were oriented along the Y axis and over the periodic boxes in a way that allowed the extremities of each filament to interact non-covalently with their periodic images, forming a continuous and stable structure.

Two stages of equilibration were performed employing the NVT and NPT ensembles each for 10 ns while keeping the protein beads restrained but allowing the water and ions to relax. Six independent production simulations (three for parallel and three for the antiparallel doublet), each with a duration of 500 ns were then performed at 313 K using the velocity rescale thermostat[65] with a coupling constant of τ = 1. Pressure coupling was implemented with the Parrinello-Rahman barostat[65] with a time constant of 12 ps. Electrostatics were treated with the reaction-field method, with

dielectric constants of 15 and infinity for charge screening in both short- and long-range regimes, respectively. The cut-off for non-bonded and electrostatic interactions in the short-range was 1.1 nm. Starting random velocities were modified at the beginning of each replicate to improve conformational sampling. Molecules were manipulated, visualized, and analysed utilizing VMD[66] and Pymol[67] software. Quantification of the interaction between filaments was performed by calculating the time any residue from any of the filaments was within a distance of 0.6 nm from each other. The interaction time of each binding-pair of residues was converted to a scale of 0 to 1, where 0 = 0% and 1 = 100% of time spent within the cut-off. For the anti-parallel fibre model, the same procedure was followed, but extra equilibration time was added to achieve a stable configuration.

### Alphafold 2 prediction of a TapA/TasA complex
The TapA/TasA complex was predicted using the Google Colab implementation of AlphaFold 2.1[35], employing multimer prediction. Signal peptide cleavage for TapA was predicted with SignalP[68]; mature sequences were used for structural predictions. Sequence alignment of the mature forms of TapA and TasA was performed using the Clustal Omega service provided by EMBL-EBI[69].

### Data visualisation and quantification
Cryo-EM images were visualised in IMOD. Fiji[70] was used for bandpass and Gaussian filtering, followed by automatic contrast adjustment. Atomic structures and tomographic data were displayed in ChimeraX. To analyse the effect of amino acid residues 28–38 truncation on fibre formation, a random subset of 20 micrographs was chosen using the *shuf* functionality of GNU, and TasA fibres were counted manually in each micrograph. For quantifying bundling in the TasA 174–177$_{AAAA}$ mutant, fibres with the same periodicity as the unaltered TasA fibres were counted and considered bundling if there were at least two visible periodic fibre interactions (see Fig. 4b, c); 100 fibres were counted from randomly selected micrographs. For homology modelling of CalY1 (Fig. S3), SWISS-MODEL[71] server was used with the cryo-EM structure of the TasA fibre as the template model.

### Statistics and reproducibility
The micrographs of WT TasA shown in Figs. 1a and S7c are representative of a dataset of 4314 micrographs that were used for structural solution of TasA (see Table S1). The tomographic reconstructions of TasA bundles shown in Figs. 3, 4 and S5 are representative of a dataset of 18 tomograms. Micrographs of mutant TasA shown in Fig. S7c (middle and right) are representative of datasets of 1317 (Δ28–38 mutant) and 4138 micrographs (174–177$_{AAAA}$ mutant). Micrographs of *B. subtilis* Δ*eps* Δ*sinR* biofilms shown in Figs. 5e–h and S9 are representative of a dataset of 362 micrographs. All micrographs were acquired using automated acquisition. In agreement with typical cryo-EM workflows, all micrographs were respectively acquired from one cryo-EM grid. AFM images of WT biofilms shown in Figs. 5e and S8a represent scans from three different biological repeats, and 26 different areas. AFM images of Δ*tasA* mutant biofilms shown in Figs. 5f and S8b represent scans from two different biological repeats and eight different areas. AFM images of Δ*sinR* Δ*eps* mutant biofilms shown in Fig. S8e, f represent scans from two different biological repeats, and 10 different areas. AFM images of TasA fibers in Fig. S8c, d represent scans from at least 30 different areas protein from at least six different purification batches.

### Reporting summary
Further information on research design is available in the Nature Portfolio Reporting Summary linked to this article.

## Data availability
The cryo-EM density map of a TasA fibre generated in this study has been deposited in the Electron Microscopy Databank (EMDB) under the accession number EMD-15673. The corresponding atomic coordinates are deposited in the Protein Data Bank (PDB) under accession code 8AUR [https://doi.org/10.2210/pdb8aur/pdb]. The atomic coordinates of monomeric TasA used for model building and structural comparison are available in the PDB under accession code 5OF1 [https://doi.org/10.2210/pdb5of1/pdb] and 5OF2 [https://doi.org/10.2210/pdb5of2/pdb]. The atomic coordinates of proteins used for structural comparison with TasA in Fig. S3 are available in the PDB under accession codes 5FLU [https://doi.org/10.2210/pdb5flu/pdb] (PapA), 6KMF [https://doi.org/10.2210/pdb6kmf/pdb] (FimA) and 4UNT [https://doi.org/10.2210/pdb4unt/pdb] (Mcg). All other data are available from the corresponding authors upon request.

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

## Acknowledgements

The authors would like to thank Sjors Scheres and Jan Löwe for helpful discussions and are grateful to Sigal Ben-Yehuda and to Bing Zhou for insightful discussions and for their generous help with strain construction. We are thankful for technical support by Shraddha Nayak at the Visual Aids department of the MRC Laboratory of Molecular Biology. L.C. and M.G. are grateful to Tsafi Danieli and Noa Dekel for fruitful discussions. T.A.M.B. is a recipient of a Sir Henry Dale Fellowship, jointly funded by the Wellcome Trust and the Royal Society (202231/Z/16/Z). T.A.M.B. would like to thank the Vallee Research Foundation, the Leverhulme Trust and the Lister Institute of Preventative Medicine for support. J.B. is supported by a Medical Research Council graduate studentship (grant numbers MR/K501256/1 and MR/N013468/1). M.G. acknowledges the support of the Neubauer Foundation for the PhD fellowship. L.C. and T.A.M.B. thank the support of the HUJI-UK-Spine joint seed funding.

## Author contributions

J.B., M.G., L.C. and T.A.M.B. designed research. J.B., M.G., C.P. and T.A.M.B. performed research J.B., M.G., C.P., D.A., S.K., L.C. and T.A.M.B. analysed data. J.B., M.G., L.C. and T.A.M.B wrote the manuscript with support from all the authors.

## Competing interests

The authors declare no competing interests.
