## [Peer Review File · Nature Communications]

Donor-strand exchange drives assembly of the TasA scaffold in *Bacillus subtilis* biofilmsReviewer #1 (Remarks to the Author):

“Molecular architecture of the TasA scaffold in *Bacillus subtilis* biofilms”

Böhning et al. present an atomic model of the biofilm matrix fibre TasA as determined by cryo-EM. They show that the fibres isolated have the same structure as those of biofilm samples using cryo-electron tomography. The fibres were found to be formed by donor-strand exchange like some bacterial pili. These results are important to the field and settle the ongoing controversy over whether these biofilm fibres were amyloid in character or not. The authors present well founded evidence that the TasA fibres, although β -rich, are non-amylogenic. We believe that this work is of great value to the fields of biofilm and protein fibre research.

Overall, the manuscript is well written with helpful, well-constructed figures. Below we have outlined some possible improvements and suggestions to improve the readability of the manuscript but believe that no further experiments are required for this story.

Comments:

Suggest hyphenating “donor-strand” as this is commonly done in literature

Line 50-53. The way this sentence is worded suggests that biofilms are surface attached communities, but the definition is broader than this and includes cell aggregates in solution for instance.

Line 70. No need to write out *Bacillus subtilis* anymore.

Line 88. Suggest adding that the X-ray diffraction studies were on in situ TasA fibres. Otherwise, the paper referenced in the previous line also showed that there was no cross- β pattern in the X-ray diffraction of TasA fibrils.

Line 88. It might be more accurate to say that the fibrils “produce a weak cross- β -sheet X-ray pattern” rather than “possess”

Line 106-107 Remove statements like “to our surprise” – this is subjective and do not need to be included in the reporting.

Line 144. Simplify this description please. We are suggesting you consider these changes: “In the fibre form, the previously self-complementing β -strand in monomeric form is displaced by the donor-strand of the (n-1)th subunit. The displaced strand folds over the opposite β -sheet. Correspondingly, with residues 39-47 are extended towards the next (n+1)th subunit, presenting the donor strand (residues 28-38), allowing the next subunit to assemble (Figure 2a).

Line 176. Are you suggesting that all the strands are parallel (no anti-parallel arrangement)? Could you state this more clearly if this is so?

Line 183. Sentence starts with “Our fits into...” and this seems to be missing a word.

Does the 174-177 alanine variant suffer from any stability issues? It looks like there is less protein isolated from the biofilm according to the western blot. Can you report # of 174-177AAAA fibres in figure S6 f?

Lines 230-232 and discussion: The complementarity of eps and TasA has been proposed in the past. Papers include:

**Collapse of genetic division of labour and evolution of autonomy in pellicle biofilms
Dragoš A, Martin M, Garcia CF, Kricks L, Pausch P, Heimerl T, Bálint B, Maróti G, Bange G, López D, Lieleg O, Kovács ÁT. Nat Microbiol. 2019 Feb;4(2):376. doi: 10.1038/s41564-**

018-0350-0.PMID: 30635640; and Erskine, E., Morris, R. J., Schor, M., Earl, C., Gillespie, R., Bromley, K. M., Sukhodub, T., Clark, L., Fyfe, P. K., Serpell, L. C., Stanley-Wall, N. R., & MacPhee, C. E. (2018). Formation of functional, non-amyloidogenic fibres by recombinant *Bacillus subtilis* TasA. *Molecular microbiology*, 110(6), 897–913. <https://doi.org/10.1111/mmi.13985> however at the level of strains being able to cross complementing each other there are several other papers. We suggest you remove this concluding sentence or you will need to ground your conclusion more firmly in the previous literature.

Line 277- format of eps needs amending

Lines 379, 396, 446 and Table S3 (perhaps other places as well)– Typo: 3610 should be 3610 if this is referring to NCIB 3610

Methods: Add in % w/v or v/v as needed in the methods.

Figure 5. Missing scale bars for pellicles and AFM images. Are the AFM images also done using height mode? Seems like there should be a height scale for each as is done in Figure S7. Why was the 'WT' strain used for AFM a TasA-mCherry strain? There is also no reference to the creation of this strain. Was it produced for this study? Is TasA still functional? Is the fusion stable and how does the mCherry integrate into the structure of the fibre?

Figure 6. Although it is only an illustration, panel C is somewhat inaccurate and thus loses usefulness. The cells are not proportional to the fibres as evident from the EM images in the paper. The cells are spaced much differently than seen in the EM images as well. Perhaps a model that looks more akin to the data would be a suitable replacement. Moreover, do you have a perspective on if the fibres are anchored to the cell body or mainly released based on your imaging? It was unclear if the cells that were not covered or touching the TasA fibres were supposed to not be producers and not coated or if the TasA fibres are secreted fully.

Table S2. Should add the reference for the paper in which each strain was created.

Figure S2 contains a panel about a possible divalent metal cation but this is not mentioned in the text at all.

Figure S4 – Would be helpful to add residue numbers for the sequence alignment.

Figure S5 – Can you indicate which loop is which, perhaps with lines/arrows and number ranges above them?

Figure S6. There is no length given for the scale bars in panel D.

Comment about data accessibility-

The accessibility of the raw data and datasets in the paper is currently challenging. The models of protein structures and images should be submitted to repositories. It is also not clear which strains the authors are referring to at time. Please link to the strain table more explicitly in figure legends and methods etc.

In many places there is no reference to how many biological and technical replicates have been completed. Examples: pellicles, colony biofilms, AFM samples... Where these only completed once or are they representative images of a dataset?

Nicola Stanley-Wall and Natalie Bamford

Reviewer #2 (Remarks to the Author):

The manuscript by Böhning et al. describes the molecular architecture of TasA, a fibre forming protein produced by the soil bacterium *Bacillus subtilis*. Importantly, TasA is a major protein component of the extracellular matrix in *B. subtilis* biofilms, and it contributes to the structure and rigidity of these multicellular assemblies. Previous studies reported a high-resolution X-ray structure of monomeric TasA and it has been speculated that TasA forms filaments with characteristics of functional amyloids. Another study, however, suggested that TasA fibres consist of a linear arrangement of globular subunits, without structural rearrangements between the monomeric and filamentous form of TasA.

Böhning et al. applied helical reconstruction from cryoEM images of purified TasA filaments to address this ambiguity. The authors reached a resolution of sub-4Å and were able to show that TasA filaments were held together by an unexpected N-terminal beta-strand exchange. These subunit interactions are reminiscent to what was observed in other bacterial filaments, such as type I pili, or other eukaryotic filaments, like uromodulin. This is a major new insight, of broad interest and beautifully described!

The authors further set out to analyze the interaction between TasA fibres by using a combination of cryo-electron tomography, atomic force microscopy as well as by the mutation of specific residues. This led to the identification of specific interaction sites involved in filament-bundling and the analysis of the created mutants revealed defects in biofilm formation.

While the results on the structure and assembly of TasA filaments are impressive and of high quality, this referee is not yet fully convinced about the second major part of this manuscript, covering the analysis of the interaction between TasA filaments *in vitro* and in biofilms. The following major comments should be addressed before publication:

Major comments:

- This reviewer is not an expert in phase-separation, however, the authors did not manage to convincingly show that TasA fibres spontaneously associate into phase-separated bundles, as stated in the title of Figure 3 and in line 175-177. What is the difference between protein aggregation into highly-ordered filament bundles and phase separation? What are the main characteristics to call this phase separation? The authors should discuss these concerns in the manuscript. Furthermore, if the authors resist on making this statement, they should include accompanying experiments, such as demonstrating concentration dependence of filament bundling. Does the assembly of bundles from different stock concentrations result in the same phase separation/filament bundling? If I understood correctly, a nematic phase is a meta state between a crystal and an isotropic liquid. Would a fourier transform of a projection image of the crystal-like bundles show discrete spots?

- The analysis of the interaction sites between a TasA doublet using atomic model fitting into a noisy tomogram density is flawed and should be removed or supported by a subtomogram average of the filament doublet. If you look closely in Figure 4c, you can even see that the fit of the lower filament, at the filament contact site right of the black box is off. This flawed fitting of the atomic structure is not even needed, as the geometrical considerations shown in Figure 4e nicely demonstrates that the loop with residues 174-177 resides furthest away from the helical centre, indicating potential interaction sites, which are further supported by the MD simulations. On top of that, the interaction between filaments in bundles seem to be different compared to the one in doublets. Whereas in doublets, contact sites only exist in the periphery, filaments nicely align along their long axis in bundles. How can this be

explained? Would it be possible to calculate a subtomogram average of bundles to visualize the three-dimensional arrangement and compare it to the interaction of filaments observed in doublets?

- The expression levels of TasA in the mutants *tasA* Δ 174-177 and *tasA* Δ 28-38 seem to be significantly reduced compared to wild-type when looking at the SDS page and Western blot in Figure S6b,c. Might this be a reason for the observed phenotypes in biofilm and pellicle formation? This should be discussed in the manuscript. Is it possible that these mutants never reach the required TasA concentration to form bundles? The cryoEM images in Figure S6d are of such poor quality in the provided PDF file, that it is impossible to properly proof the authors conclusion, especially in their example of *tasA* Δ 28-38. Here, some filaments are observable, but it seems to be at a different scale? I could not find any statement on the length of the scale bars in this panel.

- Figure 5e and f shows AFM images of WT and Δ *tasA*. While a clear difference in the biofilm architecture between these two strains is observable, the AFM images of the Δ *tasA* mutant still show some filamentous structures. Can the authors comment on the potential identity of these filaments? Besides the lower abundance of fibre-like structures, it seems to me that also several other potential ECM elements are absent in the Δ *tasA* mutant (vesicles, etc.). As the authors already have a mutant which is deficient in the production of exopolysaccharides (*B. subtilis* Δ *sinR* Δ *eps*), they could perform AFM with this mutant strain to further highlight which of the observed elements in their AFM images are indeed representing TasA fibres.

- The cryoEM images of the *B. subtilis* Δ *sinR* Δ *eps* biofilms in Figure 5g,h and Figure S8 are of rather poor quality. Why did the authors not apply cryoET for visualization? This could even highlight potential interactions between TasA fibres and *B. subtilis* cells. Would it be possible to use cryo-focused ion beam milling to thin unperturbed biofilms and to subsequently perform cryoET of biofilms in a more native state?

Minor comments:

- Figure 1a: The cryoEM image is very pixelated and it is unclear if the data is of this poor quality or if it is an artifact of PDF compression. A higher resolution image with a larger field of view together with a Zoom in would allow for a proper quality assessment of the raw data.

- Figure S2c: This panel with the putative metal ion is neither cited nor discussed in the text.

- Figure 2a,c: The indication of the N- and C-terminus in the structure would facilitate a quicker understanding of this figure.

- Figure 2b: The schematic is hard to understand, due to the two grey boxes. This could be understood as two TasA subunits, however, in my understanding it should represent one TasA monomer with a schematic of the donor strand exchange.

- Figure S2d, Figure S4: The authors frequently jump between different panels of Figure S2 and S3 which makes it hard to follow. Can the panel S2d be included in Figure S4?

- Figure 3: Similar to Fig. 1a, the provided images are pixelated, however, in movie S3 the quality of the data seems to be good. The arrow pointing to the fibre bundle in 3b seems to be a bit shifted.

- Figure 4: The arrow pointing to the doublet seems to be off target and a bit shifted.

- Figure S5: Please indicate residue numbers in your model.

- Line 207 and Figure S6d: It seems that the authors collected cryoEM images of TasA fibres (wt and mutants) which was purified from *B. subtilis* Δ sinR Δ eps mutants. This should be clearly stated in the manuscript as well as in the figure panel and not only in the legend and methods part.

- Line 213: The statement on the observation of TasA in AFM images should be toned down. e.g.: we observed filament networks potentially formed by TasA fibres adherend to cells that are not present in Δ tasA biofilms.

- Figure S7: The color of scale bars could be changed to white.

- Figure 5h: The name of the two 2D classes "This sample" and "Donor-strand TasA" might be inaccurate and should be changed to a more descriptive title.

Reviewer #3 (Remarks to the Author):

The manuscript "Molecular architecture of the TasA scaffold in *Bacillus subtilis* biofilms" by Bohning, Ghrayeb, Pedebos, Abbas, Khalid, Chai and Bharat, describes the fibres forming biofilms and built from the assembly of TasA monomers. Cryo-EM puts in evidence a new way of assembling through donor strand complementation.

The manuscript content is very interesting and deserves certainly publication. Concerning the coarse-grained molecular dynamics simulations, I was a bit puzzled by the short lengths of the trajectories. Indeed, looking in the literature, coarse-grained molecular dynamics often represent several microseconds or tenths of microseconds. Maybe, the big size (10 millions atoms) of the considered system prevents the authors to record such long trajectories, but I would strongly suggest that they extend the length of their simulations and also record some few more.

Another remark concerning the trajectories is that they are almost not analyzed. Classical analyses as coordinate RMSD could be included in the SI. It would be interesting to know how large the relative positions of TasA monomers deform during the trajectories. Also, the authors point out interacting residues belonging to different fibres (Figure S5), but do not tell much how the interactions take place: are they direct interactions, or mediated by ions or water molecules? Are the interactions established between different fibres or within the same fibre? Do they residue involved in interaction play a role in the establishment of biofilms, or have another functional role or are conserved in the sequence?

Reviewer #1 (Remarks to the Author)”

“Molecular architecture of the TasA scaffold in *Bacillus subtilis* biofilms”

Böhning et al. present an atomic model of the biofilm matrix fibre TasA as determined by cryo-EM. They show that the fibres isolated have the same structure as those of biofilm samples using cryo-electron tomography. The fibres were found to be formed by donor-strand exchange like some bacterial pili. These results are important to the field and settle the ongoing controversy over whether these biofilm fibres were amyloid in character or not. The authors present well founded evidence that the TasA fibres, although β -rich, are non-amylogenic. We believe that this work is of great value to the fields of biofilm and protein fibre research.

Overall, the manuscript is well written with helpful, well-constructed figures. Below we have outlined some possible improvements and suggestions to improve the readability of the manuscript but believe that no further experiments are required for this story.

We thank the reviewers for their helpful comments.

Comments:

Suggest hyphenating “donor-strand” as this is commonly done in literature

We have implemented this change as suggested.

Line 50-53. The way this sentence is worded suggests that biofilms are surface attached communities, but the definition is broader than this and includes cell aggregates in solution for instance.

We have adjusted the wording of the sentence to address your valid point:

L51: “Biofilms form in environmental settings as well as inside the bodies of living organisms during infections, and are further commonly found on abiotic surfaces such as medical devices.”

Line 70. No need to write out *Bacillus subtilis* anymore.

Thank you – we have adjusted this.

Line 88. Suggest adding that the X-ray diffraction studies were on in situ TasA fibres. Otherwise, the paper referenced in the previous line also showed that there was no cross- β pattern in the X-ray diffraction of TasA fibrils.

We have added this.

L88: “Moreover, a recent X-ray diffraction study showed that native TasA fibrils only produce a weak cross- β -sheet pattern *in vitro* and *in situ*.”

Line 88. It might be more accurate to say that the fibrils “produce a weak cross- β -sheet X-ray pattern” rather than “possess”

We agree that this is more accurate and have implemented this wording.

Line 106-107 Remove statements like “to our surprise” – this is subjective and do not need to be included in the reporting.

Removed as suggested.

Line 144. Simplify this description please. We are suggesting you consider these changes: “In the fibre form, the previously self-complementing β -strand in monomeric form is displaced by the donor-strand of the (n-1)th subunit. The displaced strand folds over the opposite β -sheet. Correspondingly, residues 39-47 are extended towards the next (n+1)th subunit, presenting the donor strand (residues 28-38), allowing the next subunit to assemble (Figure 2a).

We have simplified this description as follows:

L147: “In the fibre form, the previously self-complementing β -strand (residues 48-61) is displaced by the donor-strand of the (n-1)th subunit (residues 28-38). The previously self-complementing strand then folds over the opposite β -sheet, and the donor-strand (residues 28-38), along with residues 39-47, extends towards the next (n+1)th subunit, allowing assembly of the next subunit (Figure 2a).”

Line 176. Are you suggesting that all the strands are parallel (no anti-parallel arrangement)? Could you state this more clearly if this is so?

This is a very good question. The resolution of our tomograms does not allow us to unambiguously assign polarities. To answer this question, we have attempted subtomogram averaging (STA) of fibres in bundles, which we now include in the revised manuscript as Figure S5b. Unfortunately, STA of fibres in bundles did not allow us to resolve interactions between fibres, likely due to their dynamic and flexible nature. Fourier transforms (power spectra shown) of bundles reveal a wide range of spots (revised Figure S5a), indicating that fibre-fibre interactions are not rigid.

To probe this further, we have performed MD simulations for antiparallel fibres interactions, which also produced a stable solution, presented in the revised Figure S6, showing how the outer loop residues help in this interaction.

After submission of this manuscript, another study was published by Ed Egelman’s lab¹ showing that, in TasA-like fibres from hyperthermophilic archaea (termed ‘archaeal bundling pili’, ABP), arrange themselves in five parallel strands that interact with one antiparallel strand to form distinct, six-fibre bundles. There is significant structural similarity between TasA and ABP fibres, including a donor-strand exchange. Compared to TasA, however, ABP fibres form ordered, six-fibre bundles. In TasA, however, bundles of various sizes are seen, and the assembly is less ordered. We discuss this within the manuscript now.

To summarise, we think that the interaction of fibres could be either parallel or anti-parallel. TasA filaments from different cells within the biofilm might stack through

parallel or anti-parallel interactions, embedding the cells within the matrix. We have now added the above information into the paper:

L266: “We were not able to resolve distinct interactions between fibres using STA, which, in addition to lack of distinct lattice spots in Fourier transforms of bundles, suggests that interactions between fibres are not highly ordered. This interaction appears to be qualitatively similar to formation of tactoids by the phage Pf4 in *P. aeruginosa* biofilms². Nevertheless, all our experiments show the importance of a loop (containing residues 174-177) extending away from the fibre surface in mediating these interactions. Interestingly, a system was recently described in hyperthermophilic archaea (ABP pilus system), with significant structural and sequence homology to TasA¹. The ABP filaments further formed ordered, six-fibre bundles of similar morphology as TasA. While highly ordered interactions appear to exist within the ABP bundles, we could not detect a fixed size for TasA bundles, which appear to be more flexible. These stronger interactions in ABP filaments may be required as the archaea containing ABP live in acidic and boiling waters. The exact mode of interaction between TasA fibres at the atomic structural level remains to be determined, however our MD simulations suggest that both antiparallel and parallel TasA bundling could be possible.”

Line 183. Sentence starts with “Our fits into...” and this seems to be missing a word.

‘Fit’ was meant as a noun here – we, however, realise the potential for confusion. Given a comment by reviewer #2, we have removed this sentence completely.

Does the 174-177 alanine variant suffer from any stability issues? It looks like there is less protein isolated from the biofilm according to the western blot. Can you report # of 174-177AAAA fibres in figure S6 f?

We thank the reviewer for pointing this out, which we realise was not very clear in the original manuscript. The SDS-PAGE and Western blot images were originally added to verify the presence of TasA in our preparations. However, these gels were not quantitative as they showed different amounts of protein from different purifications, where concentrations were not normalised. To account for the expression levels of TasA in the wild-type and mutant strains, we now show an SDS-PAGE gel with protein preparations after the first purification step from the same number of cells (revised Figure S7b). To compare the fibre properties *in vitro* by wild-type and mutant proteins, we used the same initial concentrations. Here, TasA appears to be present in approximately equivalent amounts.

Interestingly, despite the similar concentrations used, we did detect lower numbers of fibres formed in purifications in the 174-177_{AAAA} mutant compared to the wild-type (Figure S7c), and those fibres bundled at a lower percentage compared to wild-type. This information is now also plotted in Figure S7d. It may well be that the 174-177_{AAAA} mutant fibres are slightly less stable – we discuss this now:

L201: “Also, cryo-EM of the 174-177_{AAAA} mutant fibres showed markedly reduced bundling (Figure S7c-d, 24% wild-type versus 2% for the mutant); however, this correlated with a smaller amount of TasA fibres formed by the mutant, as indicated by less fibres being detectable in the sample overall, despite the use of the same concentration of TasA.”

Lines 230-232 and discussion: The complementarity of eps and TasA has been proposed in the past. Papers include:

Collapse of genetic division of labour and evolution of autonomy in pellicle biofilms Dragoš A, Martin M, Garcia CF, Kricks L, Pausch P, Heimerl T, Bálint B, Maróti G, Bange G, López D, Lieleg O, Kovács ÁT. Nat Microbiol. 2019 Feb;4(2):376. doi: 10.1038/s41564-018-0350-0.PMID: 30635640; and

Erskine, E., Morris, R. J., Schor, M., Earl, C., Gillespie, R., Bromley, K. M., Sukhodub, T., Clark, L., Fyfe, P. K., Serpell, L. C., Stanley-Wall, N. R., & MacPhee, C. E. (2018). Formation of functional, non-amyloidogenic fibres by recombinant *Bacillus subtilis* TasA. *Molecular microbiology*, 110(6), 897–913. <https://doi.org/10.1111/mmi.13985>

however at the level of strains being able to cross complementing each other there are several other papers. We suggest you remove this concluding sentence or you will need to ground your conclusion more firmly in the previous literature.

Thank you – we have removed this sentence as suggested in the main text and cited the two papers in the discussion.

Line 277- format of eps needs amending

We have amended this.

Lines 379, 396, 446 and Table S3 (perhaps other places as well)– Typo: 361O should be 3610 if this is referring to NCIB 3610

We have fixed this – our apologies.

Methods: Add in % w/v or v/v as needed in the methods.

We have added this.

Figure 5. Missing scale bars for pellicles and AFM images. Are the AFM images also done using height mode? Seems like there should be a height scale for each as is done in Figure S7. Why was the 'WT' strain used for AFM a TasA-mCherry strain? There is also no reference to the creation of this strain. Was it produced for this study? Is TasA still functional? Is the fusion stable and how does the mCherry integrate into the structure of the fibre?

We have added scale bars in the pellicle images.

Regarding AFM images: AFM images can be taken using different modes. The most conventional mode is height mode but, when the purpose of the images is to show qualitatively the presence of certain features, it is much more informative to use 'peak force error' mode. Images taken in the latter mode do not have a real height scale and therefore it is not presented in Figure 5. To provide information about the height scale, we added into the revised Figure S8, the same AFM images that are shown in Figure 5, that were taken simultaneously in height mode, and hence they include a height scale. Please note that details observed in the fibres taken in peak force error mode

are not as apparent in these height images and this is why we chose 'peak force error' mode images for the main text and figures. We have added an explanation to the choice of AFM modes in the experimental section-

L509: “Peak force error images do not provide information on height and therefore they are presented without a height scale. However, with each peak force error image we also present the corresponding height sensor image in the extended data.”

Regarding the use of a TasA-mCherry strain: we employ a fluorescent strain as it allows us to quickly confirm via fluorescence microscopy whether TasA is appropriately expressed. The used strain was constructed for a different study³ and the ZK strain collection number (ZK5041) was specified in Table S3. We have now added a reference as well. We further add supporting evidence – showing the fluorescence microscopy images of cells within the biofilm (Figure S8g-i) and the biofilm itself compared to a wild-type (Figure S8j-k) – showing that the wildtype phenotype is retained in the TasA-mCherry strain.

Figure 6. Although it is only an illustration, panel C is somewhat inaccurate and thus loses usefulness. The cells are not proportional to the fibres as evident from the EM images in the paper. The cells are spaced much differently than seen in the EM images as well. Perhaps a model that looks more akin to the data would be a suitable replacement. Moreover, do you have a perspective on if the fibres are anchored to the cell body or mainly released based on your imaging? It was unclear if the cells that were not covered or touching the TasA fibres were supposed to not be producers and not coated or if the TasA fibres are secreted fully.

We have updated the figure to produce a more accurate representation of the AFM data, with denser cells and networks of TasA bundles at more accurate scale. We unfortunately have no structural data on how fibres are anchored to the cell yet – would be an important subject for future inquiries. Previous literature suggests that TasA fibres emanate from cells⁴, but we cannot comment on how cells and bundles interact with our data in this manuscript.

Table S2. Should add the reference for the paper in which each strain was created.

Table S2 outlines the primers used to design new strains that were used in this study. References were indeed missing in Table S3 and they have now been added.

Figure S2 contains a panel about a possible divalent metal cation but this is not mentioned in the text at all.

We now mention this in the text:

L78: “Interestingly, a density in the cryo-EM map that may represent a cation, coordinated by two negatively charged aspartate (Asp) residues, was observed (Figure S2c), which may be an ancestral remnant of the camelysin family of metalloproteinases, from which TasA has been suggested to derive⁵.”

Figure S4 – Would be helpful to add residue numbers for the sequence alignment.

We agree and have implemented this.

Figure S5 – Can you indicate which loop is which, perhaps with lines/arrows and number ranges above them?

We have added these into the revised Figure S6 as requested.

Figure S6. There is no length given for the scale bars in panel D.

Our apologies – this has been added to the caption in the revised Figure S7.

Comment about data accessibility-

The accessibility of the raw data and datasets in the paper is currently challenging. The models of protein structures and images should be submitted to repositories.

We fully agree – we have now deposited the atomic model and map to the PDB and EMDB respectively, with the accession numbers 8AUR and EMD-15673. We further attach the atomic model and cryo-EM map with this submission.

It is also not clear which strains the authors are referring to at time. Please link to the strain table more explicitly in figure legends and methods etc.

We have now specifically noted the strain number each time it is mentioned in the methods section and we refer to the strain list and strain numbers in the figure legends.

In many places there is no reference to how many biological and technical replicates have been completed. Examples: pellicles, colony biofilms, AFM samples... Where these only completed once or are they representative images of a dataset?

All the pellicle, biofilm and AFM images represent a dataset. We have now added the number of biological repeats and technical replicates to the methods section.

Nicola Stanley-Wall and Natalie Bamford

Reviewer #2 (Remarks to the Author):

The manuscript by Böhning et al. describes the molecular architecture of TasA, a fibre forming protein produced by the soil bacterium *Bacillus subtilis*. Importantly, TasA is a major protein component of the extracellular matrix in *B. subtilis* biofilms, and it contributes to the structure and rigidity of these multicellular assemblies. Previous studies reported a high-resolution X-ray structure of monomeric TasA and it has been speculated that TasA forms filaments with characteristics of functional amyloids. Another study, however, suggested that TasA fibres consist of a linear arrangement of globular subunits, without structural rearrangements between the monomeric and filamentous form of TasA.

Böhning et al. applied helical reconstruction from cryoEM images of purified TasA filaments to address this ambiguity. The authors reached a resolution of sub-4Å and were able to show that TasA filaments were held together by an unexpected N-terminal beta-strand exchange. These subunit interactions are reminiscent to what was observed in other bacterial filaments, such as type I pili, or other eukaryotic filaments, like uromodulin. This is a major new insight, of broad interest and beautifully described!

The authors further set out to analyze the interaction between TasA fibres by using a combination of cryo-electron tomography, atomic force microscopy as well as by the mutation of specific residues. This led to the identification of specific interaction sites involved in filament-bundling and the analysis of the created mutants revealed defects in biofilm formation.

While the results on the structure and assembly of TasA filaments are impressive and of high quality, this referee is not yet fully convinced about the second major part of this manuscript, covering the analysis of the interaction between TasA filaments in vitro and in biofilms. The following major comments should be addressed before publication:

We thank the reviewer for their helpful comments. We have incorporated the suggestions and toned down the messaging in the second half of the manuscript, clarifying what we can and cannot say given the data at hand. We also provide some additional experimental data to support our claims.

Major comments:

- This reviewer is not an expert in phase-separation, however, the authors did not manage to convincingly show that TasA fibres spontaneously associate into phase-separated bundles, as stated in the title of Figure 3 and in line 175-177. What is the difference between protein aggregation into highly-ordered filament bundles and phase separation? What are the main characteristics to call this phase separation? The authors should discuss these concerns in the manuscript. Furthermore, if the authors resist on making this statement, they should include accompanying experiments, such as demonstrating concentration dependence of filament bundling. Does the assembly of bundles from different stock concentrations result in the same phase separation/filament bundling? If I understood correctly, a nematic phase is a

meta state between a crystal and an isotropic liquid. Would a fourier transform of a projection image of the crystal-like bundles show discrete spots?

We thank the reviewer for this comment, and fully understand the reviewer's request to clarify. We termed the observed bundling of TasA fibres 'phase separation' in the original manuscript since it is highly reminiscent of the formation of liquid crystals (a form of phase separation) by the filamentous Pf4 phage, a structural component in *P. aeruginosa* biofilms². The term 'liquid crystal' would imply that the individual fibres are orientationally aligned, but positionally random, which is the definition of a nematic liquid crystalline phase, i.e., no discrete spots corresponding to a crystalline lattice seen in a Fourier transform.

We have projected tomograms of TasA bundles and performed Fourier transforms, and indeed, we rather see a wide range of indistinct signal, but no discrete spots corresponding to a higher order lattice, apart from those corresponding to the helical repeat of TasA fibres (revised Figure S5). We believe this also has implications for how rigid fibre-fibre interactions in bundles are, which we discuss in response to the reviewer's next comment.

We do agree, though, that to prove that this is an example of liquid-liquid phase separation will require detailed experiments. Since this is not really the focus of the manuscript, we will remove this statement. In the revised manuscript, we have removed any mention of 'phase separation' and merely refer to the similarities with other systems in the discussion.

L269: "This interaction appears to be qualitatively similar to formation of tactoids by the phage Pf4 in *P. aeruginosa* biofilms²."

- The analysis of the interaction sites between a TasA doublet using atomic model fitting into a noisy tomogram density is flawed and should be removed or supported by a subtomogram average of the filament doublet. If you look closely in Figure 4c, you can even see that the fit of the lower filament, at the filament contact site right of the black box is off. This flawed fitting of the atomic structure is not even needed, as the geometrical considerations shown in Figure 4e nicely demonstrates that the loop with residues 174-177 resides furthest away from the helical centre, indicating potential interaction sites, which are further supported by the MD simulations. On top of that, the interaction between filaments in bundles seem to be different compared to the one in doublets. Whereas in doublets, contact sites only exist in the periphery, filaments nicely align along their long axis in bundles. How can this be explained? Would it be possible to calculate a subtomogram average of bundles to visualize the three-dimensional arrangement and compare it to the interaction of filaments observed in doublets?

This is an important point, and we agree that the fit into the bundles is not as clear as our atomic structural data. As requested, we have performed subtomogram averaging (STA) of TasA fibres exclusively picked in the centre of bundles. Despite our best attempts, our STA refinements only resolve a single TasA fibre in the centre of the bundle, with no ordered density of fibres around them (revised Figure S5). We also tried to subclassify the data, but did not obtain any interacting fibres around the central fibre. This is consistent with the Fourier transform of the bundle (power spectra shown; Figure S5) showing no distinct repeats corresponding to an even partially crystalline lattice, and suggesting a more dynamic interaction.

In accordance with the reviewer's suggestion, we only show an overlay of our atomic TasA model with a cryo-ET slice, and refer to our data showing the loop is the most extended element from the fibre centre in the main text. Given we were not able to resolve an STA of fibres in bundles, we believe that the periodic interaction observed in doublets, which was also seen in MD, is not rigid, as one would expect in a crystal lattice; we hope that further studies can shed light on the exact nature of this interaction.

It is interesting to note that, after we submitted this manuscript and uploaded it to bioRxiv, another manuscript was published by Ed Egelman's lab showing the structure of TasA-like bundling pili in extremophilic archaea¹ (please see also our response to reviewer 1 above). The study shows that these so-called ABP pili are structurally similar to TasA, consisting of donor-strand exchanged subunits, and interact periodically within bundles, similar to what we observe for TasA fibre doublets. However, in contrast to TasA, ABP bundles distinctly consist of six fibres, and the structure of such six-pili bundles could be resolved to near-atomic resolution using helical reconstruction. We cannot resolve such fibre interactions for TasA, and neither is the size of the bundles fixed, suggesting variable or weaker interactions. As the reviewer stated, the fact that the loop containing residues 174-177 extends the farthest from the helical axis, that the residues interact in our MD simulations, and that mutating these residues has a profound effect of biofilm formation, suggests that these are key for forming filament-filament interfaces. We have updated the discussion in light of the above.

L266: "We were not able to resolve distinct interactions between fibres using STA, which, in addition to lack of distinct lattice spots in Fourier transforms of bundles, suggests that interactions between fibres are not highly ordered. This interaction appears to be qualitatively similar to formation of tactoids by the phage Pf4 in *P. aeruginosa* biofilms². Nevertheless, all our experiments show the importance of a loop (containing residues 174-177) extending away from the fibre surface in mediating these interactions. Interestingly, a system was recently described in hyperthermophilic archaea (ABP pilus system), with significant structural and sequence homology to TasA¹. The ABP filaments further formed ordered, six-fibre bundles of similar morphology as TasA. While highly ordered interactions appear to exist within the ABP bundles, we could not detect a fixed size for TasA bundles, which appear to be more flexible."

The reviewer also mentions that it appears in tomograms of TasA bundles that the filaments are aligned along their long axis rather than interacting periodically. We think this could just be a visual effect: Fibres also appear to be aligned in micrographs of ABP (Figure 1 of Wang et al.), but the ABP atomic-resolution structure reveals that fibre-fibre interactions within the bundle are purely periodic (Figure 3 of Wang et al.).

- The expression levels of TasA in the mutants *tasA* Δ 174-177 and *tasA* Δ 28-38 seem to be significantly reduced compared to wild-type when looking at the SDS page and Western blot in Figure S6b,c. Might this be a reason for the observed phenotypes in biofilm and pellicle formation? This should be discussed in the manuscript. Is it possible that these mutants never reach the required TasA concentration to form bundles?

We thank the reviewer for pointing this out, which we realise was not very clear in the original manuscript. The SDS-PAGE and Western blot images were originally added to verify the presence of TasA in our preparations. However, these gels were not quantitative as they showed different amounts of protein from different purifications where concentrations were not normalised. To account for the expression levels of TasA in the wild-type and mutant strains, we now show an SDS-PAGE gel with protein preparations after the first purification step from the same number of cells (revised Figure S7b). Here, TasA appears to be present in approximately equivalent amounts.

To compare the fibre properties *in vitro* by wild-type and mutant proteins, we used the same initial concentrations. Interestingly, we did detect lower numbers of fibres in purifications also in the 174-177_{AAAA} mutant compared to the wild-type (Figure S7c), and those fibres bundled at a lower percentage compared to wild-type. This information is now also plotted in Figure S7d. It may well be that the 174-177_{AAAA} mutant fibres are slightly less stable – we discuss this now:

L201: “Also, cryo-EM of the 174-177_{AAAA} mutant fibres showed markedly reduced bundling (Figure S7c-d, 24% wild-type versus 2% for the mutant); however, this correlated with a smaller amount of TasA fibres formed by the mutant, as indicated by less fibres being detectable in the sample overall, despite the use of the same concentration of TasA.”

The cryoEM images in Figure S6d are of such poor quality in the provided PDF file, that it is impossible to properly proof the authors conclusion, especially in their example of *tasA* Δ 28-38. Here, some filaments are observable, but it seems to be at a different scale? I could not find any statement on the length of the scale bars in this panel.

Apologies – the annotation for the scale bar has been added in the figure caption. We have now filtered the images differently to make the fibres more readily visible, but the fibres have a fairly small diameter (~4 nm) and it thus hard to achieve high contrast. The Δ 28-38 image appears to have poor contrast because there are no TasA fibres and only very small (~2 nm) contaminant fibres present; these may have been enriched as the purified sample contained very few TasA fibres. We now point this out in the image itself and have adjusted the figure caption to emphasise this:

L726: “A donor-strand mutant TasA (Δ 28-38 TasA, purified from MG1, right) shows a markedly reduced number of TasA fibres, while smaller aberrant, contaminating fibres can be faintly seen instead.”

- Figure 5e and f shows AFM images of WT and Δ *tasA*. While a clear difference in the biofilm architecture between these two strains is observable, the AFM images of the Δ *tasA* mutant still show some filamentous structures. Can the authors comment on the potential identity of these filaments? Besides the lower abundance of fibre-like structures, it seems to me that also several other potential ECM elements are absent in the Δ *tasA* mutant (vesicles, etc.). As the authors already have a mutant which is deficient in the production of exopolysaccharides (*B. subtilis* Δ *sinR* Δ *eps*), they could perform AFM with this mutant strain to further highlight which of the observed elements in their AFM images are indeed representing TasA fibres.

We thank the reviewer for this suggestion. Some of the filamentous structures present in ΔtasA may be flagella, which are abundant in *B. subtilis* biofilms and can also be seen in the cryo-EM imaging data of $\Delta\text{eps} \Delta\text{sinR}$ biofilms in Figure S9. As for the lack of vesicles etc., in the AFM images – sample preparation for AFM imaging includes several washing steps that would most likely eliminate all the elements that are not adsorbed on the mica surface. We are unsure whether the features observed are really vesicles or some other molecules in a different orientation.

As requested, we have performed AFM of $\Delta\text{eps} \Delta\text{sinR}$ pellicles. Significantly increased numbers of filamentous structures can be seen compared to ΔtasA (revised Figure S8). We added a paragraph to the manuscript:

L212: “Next, by imaging intact wild-type pellicle biofilms using atomic force microscopy (AFM), we observed networks of filaments adherent to cells, potentially formed by TasA fibres, that are severely reduced in ΔtasA biofilms (Figure 5e-f; Figure S8). Similarly, greatly increased numbers of fibres could again be seen in a strain that overproduces TasA but lacks *eps* ($\Delta\text{sinR} \Delta\text{eps}$) (Figure S8e-f).”

- The cryoEM images of the *B. subtilis* $\Delta\text{sinR} \Delta\text{eps}$ biofilms in Figure 5g,h and Figure S8 are of rather poor/quality. Why did the authors not apply cryoET for visualization? This could even highlight potential interactions between TasA fibres and *B. subtilis* cells. Would it be possible to use cryo-focused ion beam milling to thin unperturbed biofilms and to subsequently perform cryoET of biofilms in a more native state?

We have replaced potentially low-resolution images – we hope these will be an improvement. We did acquire some cryo-ET data but its appearance is overall worse than of the shown high-dose 2D images. Particularly, the *B. subtilis* cells are considerably too thick to reconstruct good tomograms without prior thinning.

We think the possibility of performing cryo-FIB milling is an excellent point and would likely be required to resolve interactions between the cell and the fibre bundles. We are looking to address this in future studies, since this would be a significant undertaking beyond the scope of this paper on the TasA atomic structure.

Minor comments:

- Figure 1a: The cryoEM image is very pixelated and it is unclear if the data is of this poor quality or if it is an artifact of PDF compression. A higher resolution image with a larger field of view together with a Zoom in would allow for a proper quality assessment of the raw data.

We have now replaced the image with a differently filtered image and hope that it appears improved to the reviewer – we think that this image is representative of the raw cryo-EM data, which was acquired relatively close to focus (between -1 and -2 μm defocus). As with cryo-EM data of small fibres (the diameter of a TasA subunit is less than 4 nm), the raw micrographs are intrinsically noisy, but we hope that the image conveys the wave-like shape of the TasA fibre.

- Figure S2c: This panel with the putative metal ion is neither cited nor discussed in the text.

We now mention it within the Results section:

L125: “Interestingly, a density in the cryo-EM map that may represent a cation, coordinated by two negatively charged aspartate (Asp) residues, was observed (Figure S2c), which may be an ancestral remnant of the camelysin family of metalloproteinases, from which TasA has been suggested to derive⁵.”

- Figure 2a,c: The indication of the N- and C-terminus in the structure would facilitate a quicker understanding of this figure.

Thank you, we agree that this makes the structural re-arrangement more easily visible. We have implemented this into the figure.

- Figure 2b: The schematic is hard to understand, due to the two grey boxes. This could be understood as two TasA subunits, however, in my understanding it should represent one TasA monomer with a schematic of the donor strand exchange.

Indeed, the grey boxes were meant to represent the individual beta-sheets. We have now adjusted this to indicate the subunit more clearly with an additional box and adjusted the description within the figure.

- Figure S2d, Figure S4: The authors frequently jump between different panels of Figure S2 and S3 which makes it hard to follow. Can the panel S2d be included in Figure S4?

We have added panel S2d to Figure S4.

- Figure 3: Similar to Fig. 1a, the provided images are pixelated, however, in movie S3 the quality of the data seems to be good.

We have replaced these with differently filtered images – the quality should be comparable to the movie now.

The arrow pointing to the fibre bundle in 3b seems to be a bit shifted.

Thanks – this is now adjusted.

- Figure 4: The arrow pointing to the doublet seems to be off target and a bit shifted.

We have adjusted this.

- Figure S5: Please indicate residue numbers in your model.

We have added the numbers into the revised Figure S6 as requested.

- Line 207 and Figure S6d: It seems that the authors collected cryoEM images of TasA fibres (wt and mutants) which was purified from *B. subtilis* Δ sinR Δ eps mutants. This should be clearly stated in the manuscript as well as in the figure panel and not only in the legend and methods part.

We now mention this in the main text and in the legend for Figure 1.

L101: “To gain insights into the structure of TasA fibres, we purified TasA from *B. subtilis* $\Delta eps \Delta sinR$ using previously established procedures⁶.”

- Line 213: The statement on the observation of TasA in AFM images should be toned down. e.g.: we observed filament networks potentially formed by TasA fibres adherend to cells that are not present in $\Delta tasA$ biofilms.

We have adjusted this statement as suggested.

L212: “Next, by imaging intact wild-type pellicle biofilms using atomic force microscopy (AFM), we observed networks of filaments adherent to cells, potentially formed by TasA fibres, that are severely reduced in $\Delta tasA$ biofilms (Figure 5e-f; Figure S8). Similarly, greatly increased numbers of fibres could again be seen in a strain that overproduces TasA but lacks *eps* ($\Delta sinR \Delta eps$) (Figure S8e-f).”

- Figure S7: The color of scale bars could be changed to white.

We have updated this (now Figure S8).

- Figure 5h: The name of the two 2D classes “This sample” and “Donor-strand TasA” might be inaccurate and should be changed to a more descriptive title.

We have updated this and now use the terms ‘biofilm sample’ and ‘*in vitro* sample’.

Reviewer #3 (Remarks to the Author):

The manuscript "Molecular architecture of the TasA scaffold in *Bacillus subtilis* biofilms" by Bohning, Ghrayeb, Pedebos, Abbas, Khalid, Chai and Bharat, describes the fibres forming biofilms and built from the assembly of TasA monomers. Cryo-EM puts in evidence a new way of assembling through donor strand complementation.

The manuscript content is very interesting and deserves certainly publication.

We thank the reviewer for their comments.

Concerning the coarse-grained molecular dynamics simulations, I was a bit puzzled by the short lengths of the trajectories. Indeed, looking in the literature, coarse-grained molecular dynamics often represent several microseconds or tenths of microseconds. Maybe, the big size (10 millions atoms) of the considered system prevents the authors to record such long trajectories, but I would strongly suggest that they extend the length of their simulations and also record some few more.

We have now extended the simulations to 500 ns and recorded triplicates – these are now shown in the revised Figure S6. The results and inferences from the simulations are the same as reported in the first version of the manuscript.

Another remark concerning the trajectories is that they are almost not analyzed. Classical analyses as coordinate RMSD could be included in the SI. It would be interesting to know how large the relative positions of TasA monomers deform during the trajectories. Also, the authors point out interacting residues belonging to different fibres (Figure S5), but do not tell much how the interactions take place: are they direct interactions, or mediated by ions or water molecules? Are the interactions established between different fibres or within the same fibre? Do they residue involved in interaction play a role in the establishment of biofilms, or have another functional role or are conserved in the sequence?

We have now annotated the image interactions further and added RMSD plots as well as incorporated RMSF data in Figure S6. The interactions are between residues of different fibres and direct protein-protein interactions (information now added to the figure caption). We believe that the outermost loop residues, which play an important role in fibre-fibre interaction are key to the establishment of biofilms, because mutating them prevents efficient biofilm formation (see Figures 5a and S7a)

References

- 1 Wang, F., Cvirkaite-Krupovic, V., Krupovic, M. & Egelman, E. H. Archaeal bundling pili of *Pyrobaculum calidifontis* reveal similarities between archaeal and bacterial biofilms. *Proceedings of the National Academy of Sciences* **119**, e2207037119 (2022).
- 2 Tarafder, A. K. *et al.* Phage liquid crystalline droplets form occlusive sheaths that encapsulate and protect infectious rod-shaped bacteria. *Proceedings of the National Academy of Sciences* **117**, 4724-4731 (2020).
- 3 Kolodkin-Gal, I. *et al.* D-amino acids trigger biofilm disassembly. *Science* **328**, 627-629 (2010).
- 4 Romero, D., Vlamakis, H., Losick, R. & Kolter, R. An accessory protein required for anchoring and assembly of amyloid fibres in *B. subtilis* biofilms. *Molecular Microbiology* **80**, 1155-1168 (2011).
- 5 Diehl, A. *et al.* Structural changes of TasA in biofilm formation of *Bacillus subtilis*. *Proceedings of the National Academy of Sciences* **115**, 3237-3242 (2018).
- 6 Chai, L. *et al.* Isolation, characterization, and aggregation of a structured bacterial matrix precursor. *Journal of Biological Chemistry* **288**, 17559-17568 (2013).

Reviewer #1 (Remarks to the Author):

The authors have addressed the points raised in the first review clearly.

A couple of minor points are listed below.

Line 72 – listing saying “biofilms and pellicles” suggests that pellicles are not a type of biofilm. Would be better to say colony biofilms.

Line 115-117 – The C-terminus was not “unresolved” per se, in the crystal structure as it wasn’t included in the construct that was crystallised.

However new data has been included -

The tasA-mCherry fusion has been used by the Bacillus subtilis biofilm field before. However no evidence has been provided in any publication that shows it is functional. To include the data relating to mCherry, you will need to provide other data e.g., immunoblot data showing that the fusion is full length and that it can functionally substitute for native tasA should be included. Alternatively, remove the images and text surrounding the fluorescence - this is the advised option.

Reviewer #2 (Remarks to the Author):

The authors have made an effort to address the concerns and the revised version of the manuscript has been improved according to the reviewers' recommendations.

Some minor comments which the authors could consider:

- line 113-114: Please add a sentence, why you use *B. subtilis* Δ eps Δ sinR and not the wild-type to purify TasA fibers. This might not be clear to the reader.

- Figure 5e/f, line 228-229 and Figure S8: Please add an arrowhead to indicate what you identify as potential TasA fibers in your AFM images. This might facilitate the understanding. For me it was hard to judge, what might be a flagellum and what a TasA fiber.

- The discussion has some repetitions (e.g. lines 359-366 and 377-379) and could be shortened.

Reviewer #3 (Remarks to the Author):

The authors have responded satisfactorily to my previous remarks and I therefore recommend publication of the manuscript.

Reviewer #1:

Line 72 – listing saying “biofilms and pellicles” suggests that pellicles are not a type of biofilm. Would be better to say colony biofilms.

We adjusted this according to the reviewers' suggestion. Please note that the line numbers should correspond to the tracked changes version submitted as .docx (the PDF conversion on the Nature Communications website changes the line numbers):

L239: “Colony biofilms and pellicles with this mutation (174-177_{AAAA}) showed an aberrant morphology compared to wild-type *B. subtilis* biofilms [...]”

Line 115-117 – The C-terminus was not “unresolved” per se, in the crystal structure as it wasn't included in the construct that was crystallised.

Indeed, this we realise this was misleading – we have adjusted this accordingly:

L122: The C-terminal residues 239-261, previously found to be unstructured in NMR studies of monomeric TasA and not present in the monomer X-ray structure [...]”

However new data has been included -

The tasA-mCherry fusion has been used by the *Bacillus subtilis* biofilm field before. However no evidence has been provided in any publication that shows it is functional. To include the data relating to mCherry, you will need to provide other data e.g., immunoblot data showing that the fusion is full length and that it can functionally substitute for native tasA should be included. Alternatively, remove the images and text surrounding the fluorescence - this is the advised option.

We thank the reviewers for this comment and have removed the recently added images accordingly from Figure S8.

Reviewer #2:

Some minor comments which the authors could consider:

- line 113-114: Please add a sentence, why you use *B. subtilis* Δ eps Δ sinR and not the wild-type to purify TasA fibers. This might not be clear to the reader.

This strain was used as it was previously shown to facilitate purification of TasA in the absence of exopolysaccharides (eps), an abundant matrix component. We now elaborate on this in the manuscript:

L107: “To gain insights into the structure of TasA fibres, we purified TasA from a *B. subtilis* double mutant strain ($\Delta sinR \Delta eps$) that lacks the matrix polysaccharide operon *eps*, thereby producing TasA that is not associated with EPS, as well as the regulator gene *sinR*, an *eps* and *tasA* repressor, causing the strain to overproduce TasA¹⁷.”

- Figure 5e/f, line 228-229 and Figure S8: Please add an arrowhead to indicate what you identify as potential TasA fibers in your AFM images. This might facilitate the understanding. For me it was hard to judge, what might be a flagellum and what a TasA fiber.

It is not trivial to determine whether a given fibre in an AFM image corresponds to a flagellum or a TasA fibre. Deletion of *tasA* results in a significant reduction in visible fibres, and we think it is likely that remaining fibres in $\Delta tasA$ correspond to flagella as these are the other major filaments in biofilm cryo-EM images (Figure S9). We realise that the caption of Figure 5 may have been misleading in this regard and adjusted it accordingly:

L426: “AFM imaging comparing *B. subtilis* pellicles of e) wild-type (ZK5041 strain, table S3) and f) $\Delta tasA$ (ZK3657 strain) shows that deletion of *tasA* results in a significant reduction in visible fibres. Fibres remaining in $\Delta tasA$ pellicles may correspond to flagella (see Figure S9).”

- The discussion has some repetitions (e.g. lines 359-366 and 377-379) and could be shortened.

We have made text adjustments in the discussion to make it more concise and prevent repetition.

We have also taken the opportunity to standardise the entire manuscript stylistically. We include a tracked change version of the manuscript for convenience. We have now also updated the spelling of ‘donor strand’ in all instances to be fully compatible with previous literature (see ref¹, ref² and ref³). This way, ‘donor-strand exchange’ and ‘donor-strand complementation’ are hyphenated, but ‘donor strand’ on its own is not.

References:

- 1 Remaut, H. *et al.* Donor-strand exchange in chaperone-assisted pilus assembly proceeds through a concerted β strand displacement mechanism. *Molecular Cell* **22**, 831-842 (2006).
- 2 Hospenthal, M. K. *et al.* Structure of a chaperone-usher pilus reveals the molecular basis of rod uncoiling. *Cell* **164**, 269-278 (2016).
- 3 Shibata, S. *et al.* Structure of polymerized type V pilin reveals assembly mechanism involving protease-mediated strand exchange. *Nature Microbiology* **5**, 830-837 (2020).